# Comparative studies of 2168 plasma proteins measured by two affinity-based platforms in 4000 Chinese adults

Baihan Wang[1], Alfred Pozarickij[1], Mohsen Mazidi[1], Neil Wright [1], Pang Yao[1], Saredo Said[1], Andri Iona[1], Christiana Kartsonaki[1], Hannah Fry [1], Kuang Lin[1], Yiping Chen [1], Huaidong Du [1], Daniel Avery [1], Dan Schmidt-Valle[1], Canqing Yu [2,3,4], Dianjianyi Sun [2,3,4], Jun Lv[2,3,4], Michael Hill [1], Liming Li [2,3,4], Derrick A. Bennett [1], Rory Collins[1], Robin G. Walters [1], Robert Clarke [1], Iona Y. Millwood [1,5] ✉ & Zhengming Chen [1,5] ✉ On behalf of China Kadoorie Biobank Collaborative Group*

Proteomics offers unique insights into human biology and drug development, but few studies have directly compared the utility of different proteomics platforms. We measured plasma levels of 2168 proteins in 3976 Chinese adults using both Olink Explore and SomaScan platforms. The correlation of protein levels between platforms was modest (median rho = 0.29), with protein abundance and data quality parameters being key factors influencing correlation. For 1694 proteins with one-to-one matched reagents, 765 Olink and 513 SomaScan proteins had *cis*-pQTLs, including 400 with colocalising *cis*-pQTLs. Moreover, 1096 Olink and 1429 SomaScan proteins were associated with BMI, while 279 and 154 proteins were associated with risk of ischaemic heart disease, respectively. Addition of Olink and SomaScan proteins to conventional risk factors for ischaemic heart disease improved C-statistics from 0.845 to 0.862 (NRI: 12.2%) and 0.863 (NRI: 16.4%), respectively. These results demonstrate the utility of these platforms and could inform the design and interpretation of future studies.

Proteins play a key role in human health and most drugs target proteins, including enzymes, antibodies, transport, or structural proteins. Measurements of plasma levels of proteins, particularly when combined with genetic and phenotypic information, can help to understand biological mechanisms and disease aetiology, improve disease risk prediction, and evaluate novel protein targets for drug treatment of specific diseases. Advances in high-throughput proteomic assays now enable the measurement of thousands of plasma proteins, and their application in population and clinical studies is likely to transform the development of precision medicine.

The relative or absolute plasma levels of proteins can be measured using different technologies, including mass spectrometry and affinity-based methods. Mass spectrometry methods identify proteins based on peptides following enzymatic digestion and have been widely used for both targeted (focusing on a few hundred pre-defined proteins) and non-targeted (measuring up to 4500 plasma proteins) approaches in clinical and population studies[1–3]. However, due to the

[1]Clinical Trial Service Unit, Nuffield Department of Population Health, University of Oxford, Oxford, UK. [2]Department of Epidemiology & Biostatistics, School of Public Health, Peking University, Beijing, China. [3]Peking University Center for Public Health and Epidemic Preparedness and Response, Beijing, China. [4]Key Laboratory of Epidemiology of Major Diseases (Peking University), Ministry of Education, Beijing, China. [5]These authors contributed equally: Iona Y. Millwood, Zhengming Chen. *A list of authors and their affiliations appears at the end of the paper. ✉e-mail: iona.millwood@ndph.ox.ac.uk; zhengming.chen@ndph.ox.ac.uk

extensive pre-fractionation required[4], application of mass spectrometry in large-scale population studies remains challenging. In contrast, advances in high-throughput affinity-based protein profiling technology using the antibody-based Olink[5] or the aptamer-based SomaScan[6–8] platform have made it possible to measure plasma levels of several thousand different protein markers simultaneously in large numbers of samples. The Olink platform measures protein levels using paired antibodies binding a single target protein[5], while SomaScan employs slow off-rate modified aptamers (SOMAmers) as protein-binding reagents[6–8].

Both Olink and SomaScan platforms have recently been used in large-scale population studies, helping to identify genetic variants associated with plasma protein levels (i.e. protein quantitative trait loci; pQTLs) and biomarkers of traits (e.g. BMI), diseases, and their progression[9–12]. The most recent Olink Explore HT and SomaScan platforms now include >5000 and >11000 protein assays, respectively, targeting a large number of overlapping proteins. A few small studies have directly compared the analytical performance of different platforms in the same sample, but these typically involved European ancestry populations and reported only modest correlations between protein levels measured by Olink and SomaScan[13–16]. Moreover, the protein-phenotype associations and pQTLs identified in these studies varied between the different platforms, as did the number of overlapping proteins[15–17]. In a recent large study, the analyses were based mainly on indirect comparisons of different individuals enroled in the UK Biobank and the deCODE study[17]. Furthermore, while most previous studies focused on the comparison of pQTLs, observational proteomic associations with phenotypes are also important for biomarker discovery and risk prediction of diseases. Therefore, more studies are needed to directly compare different assay platforms in identical samples that involve larger sample sizes, more overlapping proteins, and more comprehensive analyses from both genetic and phenotypic perspectives.

We present *direct* comparative analyses of 2168 proteins measured by both the Olink Explore 3072 (~3000 protein targets) and the SomaScan Assay v4.1 (~7000 protein targets) platforms in 3976 participants from an ischaemic heart disease (IHD) case-subcohort study within the China Kadoorie Biobank (CKB). The main aims of this study were to: (i) examine the correlations between plasma protein levels measured by the two platforms; (ii) compare the pQTLs identified in genome-wide association studies (GWAS); (iii) compare the associations of proteins with different traits (e.g. adiposity) and disease risks (e.g. IHD); and (iv) compare the performance of proteins for prediction of IHD risk, separately and in combination with conventional risk factors.

## Results

Of the 3976 participants in the CKB proteomics study (1951 incident IHD cases and 2025 subcohort participants), 53.7% were female and the mean (SD) age at baseline was 57.3 (11.7) years and the mean (SD) BMI was 23.9 (3.6) kg/m². At sample collection, the mean ambient temperature was 15.7 °C (SD 10.5), and the mean time since last meal was 4.7 (SD 4.7) hours. Further details of participant characteristics by case-subcohort ascertainment are shown in Table 1.

As shown in Fig. 1, after excluding duplicated reagents and reagents without corresponding UniProt IDs, there were 2923 reagents (mapped to 2923 UniProt IDs) included in the Olink Explore 3072 platform, and 7301 SOMAmers (mapped to 6397 UniProt IDs) included in SomaScan Assay v4.1. We identified 2749 Olink-SomaScan reagent pairs, which mapped to 2168 UniProt IDs (since some proteins were targeted by multiple SOMAmers and some SOMAmers targeted multiple proteins). This included 1694 reagent pairs for which one unique SOMAmer was matched to one unique Olink reagent.

### Observational correlations and associated factors

The SomaScan data were provided in two versions, one with adaptive normalization by maximum likelihood (ANML; a procedure to

**Table 1 | Baseline characteristics of IHD cases and subcohort participants who had no prior history of CVD at baseline**

| Characteristics[a] | IHD cases (n = 1951) | Subcohort (n = 2025) | Total (n = 3976) |
|---|---|---|---|
| **Socio-demographics** | | | |
| Mean age (SD), years | 63.7 (9.3) | 51.3 (10.5) | 57.3 (11.7) |
| Female, % | 45.1 | 62.1 | 53.7 |
| Urban residents, % | 47.0 | 50.6 | 48.8 |
| Married with spouse, % | 79.9 | 91.3 | 85.7 |
| ≥6 years education, % | 37.7 | 52.1 | 45.1 |
| **Clinical measurements, mean (SD)** | | | |
| Ambient temperature, °C | 15.4 (10.5) | 16.0 (10.5) | 15.7 (10.5) |
| Time since last meal, hours | 4.3 (4.4) | 5.1 (5.0) | 4.7 (4.7) |
| BMI, kg/m² | 24.0 (3.8) | 23.8 (3.4) | 23.9 (3.6) |
| SBP, mmHg | 146.3 (24.6) | 130.5 (21.4) | 138.3 (24.3) |
| DBP, mmHg | 80.4 (12.8) | 78.0 (11.1) | 79.2 (12.0) |
| Heart rate, bpm | 79.4 (12.5) | 78.8 (11.7) | 79.1 (12.1) |
| RBG, mmol/L | 7.2 (3.9) | 6.0 (2.3) | 6.6 (3.3) |
| **Self-reported medical history, %** | | | |
| Poor self-rated health | 14.6 | 8.3 | 11.4 |
| Diabetes[b] | 10.3 | 3.1 | 6.6 |
| Chronic kidney disease | 1.4 | 1.3 | 1.4 |
| Cancer | 0.6 | 0.6 | 0.6 |
| **Lifestyle habits** | | | |
| Physical activity (SD), MET-h/day | 13.0 (10.5) | 21.3 (14.5) | 17.3 (13.3) |
| Ever regular smoking, % | | | |
| Male | 75.1 | 74.8 | 75.0 |
| Female | 9.3 | 3.3 | 5.8 |
| Regular alcohol drinking, % | | | |
| Male | 25.0 | 36.1 | 29.6 |
| Female | 2.8 | 2.4 | 2.6 |

*CVD* cardiovascular disease, *SD* standard deviation, *BMI* body mass index, *SBP* systolic blood pressure, *DBP* diastolic blood pressure, *RBG* random blood glucose, *MET* metabolic equivalent.
[a]There were 40 participants with missing data for ambient temperature, and 31 participants with missing data for random glucose.
[b]Also included cases that were screen-detected at baseline based on random and/or fasting blood glucose levels.

normalise protein levels to an external reference to control for inter-sample variability) and one without ANML, and there were high correlations of protein levels (median Spearman's rho = 0.82) between them. The main analyses compared Olink with SomaScan-non-ANML data, as Olink did not include the ANML step, and there were higher correlations of Olink with SomaScan-non-ANML than with SomaScan-ANML data (Supplementary Fig. 1). The results based on SomaScan-ANML are also provided in Supplementary Information.

For all 2749 Olink-SomaScan reagent pairs, we found a median Spearman's rho of 0.29 between protein levels measured by two platforms (Supplementary Figs. 1, 2; Supplementary Data 1). Histograms suggest a bimodal distribution of Spearman's rho, with a peak centred around 0 and another closer to 0.8. When restricting the analysis to the 1694 one-to-one matched pairs, we found a similar distribution of Spearman's rho with a median of 0.26 (Fig. 2a; Supplementary Figs. 3, 4). For clarity, we have presented the results separately for the 1694 proteins with one-to-one matched pairs and the 474 proteins with non-one-to-one matched pairs. Analysis restricted to the 2025 randomly selected subcohort participants yielded similar

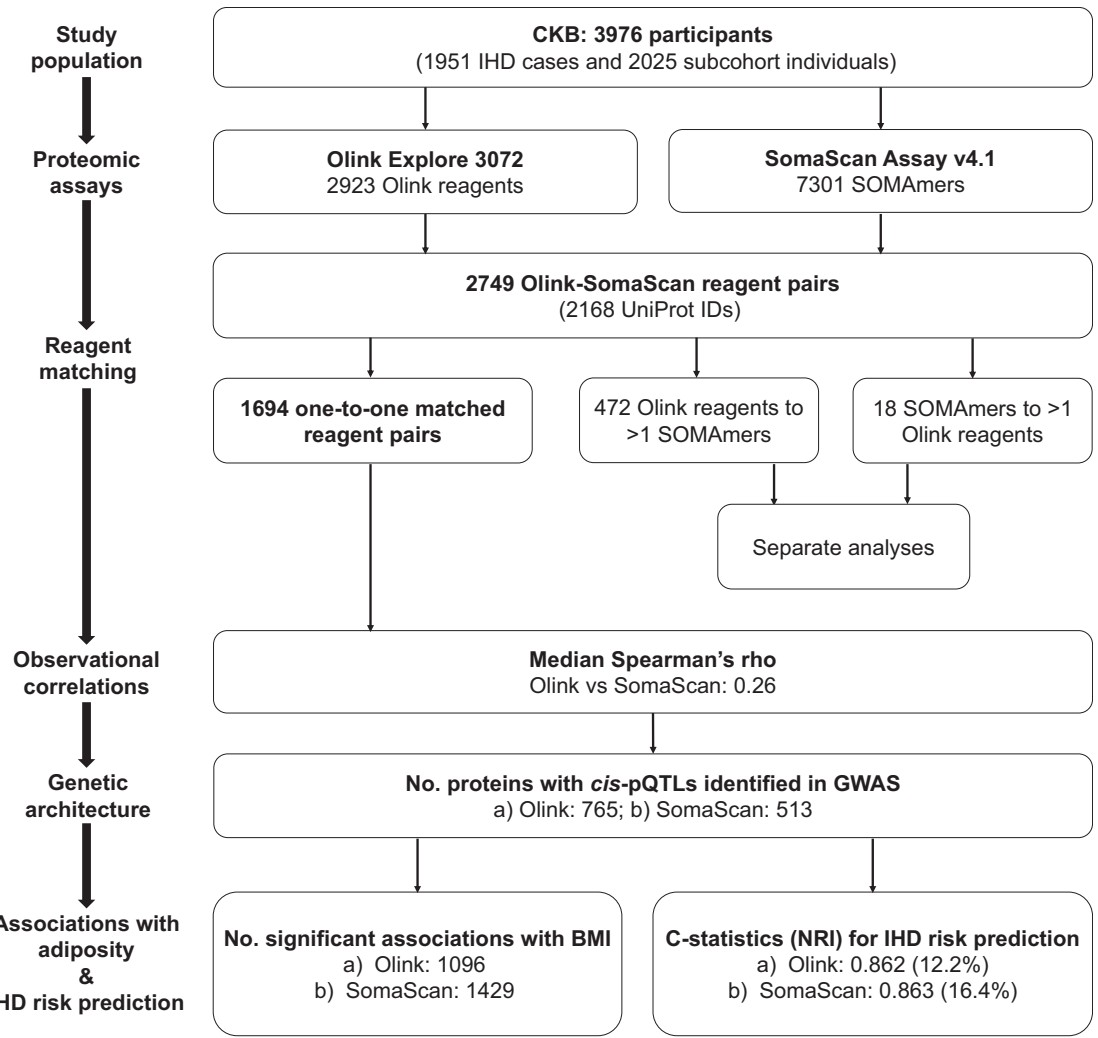

**Fig. 1 | Summary of study design, analytical approaches, and key findings.** Main analyses were conducted on 1694 one-to-one matched Olink-SomaScan reagent pairs in 3976 CKB participants. Results were corrected for multiple testing using false discovery rate within each platform. Risk prediction models for IHD were built using conventional risk factors (age, sex, smoking, type 2 diabetes, systolic blood pressure, and waist circumference) and 1694 matched proteins. CKB China Kadoorie Biobank, IHD ischaemic heart disease, NRI net reclassification index.

results (Supplementary Fig. 5). Matching of reagents based on UniProt IDs was largely consistent with matching based on observational correlations (Supplementary Note 2; Supplementary Fig. 6; Supplementary Data 2).

Using Boruta feature selection, we found several assay- and sample-related factors predictive of Spearman's correlation coefficients (Fig. 2b; Supplementary Fig. 7). Higher protein abundance (i.e. lower dilution factors; Supplementary Figs. 8, 9) and higher data quality (i.e. lower % below limit of detection [LOD] and lower % of samples with quality control [QC] warnings) were predictive of higher correlations between the platforms. Negative skewness (more values on the right side of the distribution) and platykurtic distributions (fewer extreme values, also indicated by lower % outliers) were also associated with higher correlations. Proteins in Olink batch 1 were predictive of higher correlations, which were typically higher in abundance and also assayed at a different laboratory from Olink batch 2. In contrast, protein characteristics retrieved from the UniProt Knowledgebase and GO annotations were generally not predictive of correlations (Supplementary Tables 1, 2).

## Comparisons of pQTLs

In GWAS of the 1694 reagent pairs, we identified 765 (45.2%) proteins with *cis*-pQTLs for Olink and 513 (30.3%) proteins with *cis*-pQTLs for SomaScan, corresponding to 794 and 526 sentinel *cis*-pQTL variants, respectively (Fig. 2c; Supplementary Fig. 10; Supplementary Data 3, 4). Moreover, there were 705 (41.6%) and 723 (42.7%) proteins with *trans*-pQTLs, respectively, corresponding to 1139 and 1095 sentinel *trans*-pQTL variants (Supplementary Data 3, 4).

In colocalisation analysis of 850 proteins with *cis*-pQTLs discovered in Olink and/or SomaScan, 400 (47.1%) proteins had *cis*-pQTLs that colocalised between the platforms (Fig. 2d; Supplementary Fig. 11). When restricting the analysis to 428 proteins with *cis*-pQTLs identified in *both* platforms, 333 (77.8%) colocalised between Olink and SomaScan. Proteins with colocalising *cis*-pQTLs had higher between-platform correlations (median rho = 0.72; Fig. 2a; Supplementary Fig. 12). In addition, we estimated the $r^2$ between the *cis*-pQTL sentinel variants for proteins with *cis*-pQTLs in both platforms, if the *cis*-pQTL regions overlapped between Olink and SomaScan. Of these 417 proteins (428 sentinel variant pairs), the mean $r^2$ was 0.75 between all sentinel variant pairs, and 310 proteins had at least one pair of sentinel variants with an $r^2 > 0.6$. We also identified 1050 proteins with *trans*-pQTLs discovered in Olink and/or SomaScan, of which 311 (29.6%) proteins had colocalising *trans*-pQTLs between the platforms, with a median rho = 0.76 for the observational correlations of these 311 proteins (Supplementary Data 3).

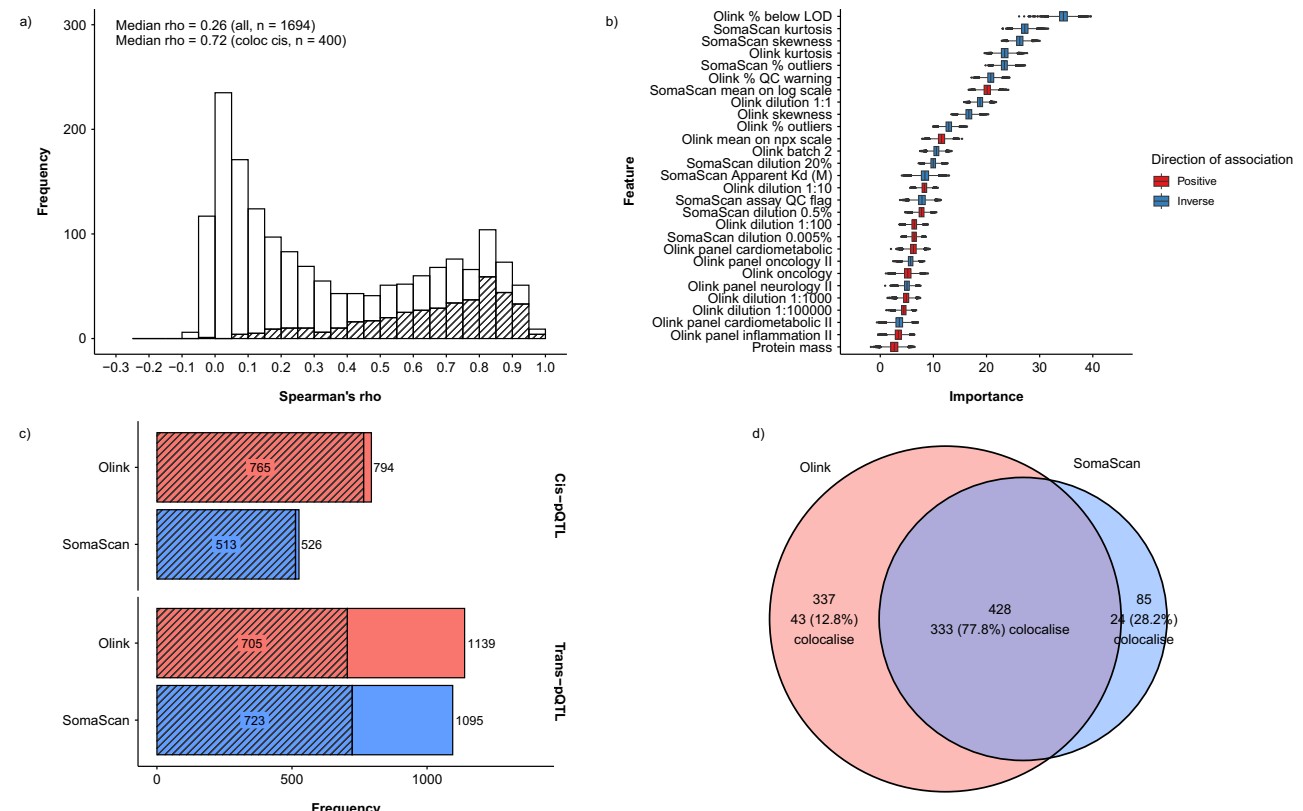

**Fig. 2 | Observational correlations, associated factors, and comparison of pQTLs for 1694 proteins measured using both Olink and SomaScan platforms.** Analyses were conducted for 1694 matched proteins among 3976 participants. **a** Observational correlations between Olink and SomaScan, with shaded areas indicating proteins with colocalising *cis*-pQTLs. **b** Features predictive of Spearman's rho and their importance in Boruta feature selection. 28 of 87 features were selected by Boruta with *p* values < 0.01 (two-sided). Importance measures represent Z-scores of mean decrease accuracy measure with normalised permutation. The centre, bounds, and whiskers of boxes represent the median, first/third quartile, and 1.5 times the interquartile range of the data. **c** Number of sentinel pQTLs identified in Olink and SomaScan, with shaded areas indicating the number of proteins with pQTLs. **d** Number of proteins with *cis*-pQTLs identified in Olink and SomaScan, and number of which had *cis*-pQTLs that colocalised between the platforms. pQTL protein quantitative trait loci.

For each platform, we classified the *cis*-pQTLs based on whether their sentinel variants were protein-altering variants (PAVs) or in linkage equilibrium (LD; $r^2 > 0.8$) with PAVs (i.e. PAV *cis*-pQTLs). We identified 319 (40.2%) PAV *cis*-pQTL variants for 319 Olink proteins and 219 (41.7%) PAV *cis*-pQTL for 219 SomaScan proteins. Proteins with PAV *cis*-pQTLs had similar observational correlations (median rho = 0.49 for Olink and 0.66 for SomaScan) to those with non-PAV *cis*-pQTLs (median rho = 0.53 for Olink and 0.71 for SomaScan). 148 cis-pQTLs were PAVs or in LD with PAVs in both Olink and SomaScan, of which 122 (82.4%) colocalised. To identify factors that might explain the discordant findings on *cis*-pQTLs between Olink and SomaScan, we then compared whether having a PAV *cis*-pQTL, as well as observational correlations and technical factors, were associated with whether or not a protein had colocalising *cis*-pQTLs between the platforms. In general, proteins with lower observational correlations, proteins of lower abundance, and proteins with more samples below LOD in Olink were more likely to have platform-specific *cis*-pQTLs. Proteins with lower observational correlations were also more likely to have non-colocalising *cis*-pQTLs identified in both platforms. Compared to SomaScan non-PAV *cis*-pQTLs, SomaScan PAV *cis*-pQTLs were more likely to not colocalise with Olink *cis*-pQTLs identified for the same protein (Supplementary Fig. 13).

In this East Asian population, we identified pQTLs that had not been previously reported in other populations due to minor allele frequency (MAF) differences. For example, the *cis*-pQTL sentinel variant (rs4646776) for ALDH2 identified in both platforms has a MAF =

0.22 in East Asians but <0.001 in other populations, and is in high LD with a known functional variant rs671 that affects alcohol metabolism (Supplementary Fig. 14; Supplementary Data 4)[18]. Moreover, the *cis*-pQTL sentinel variant (rs76863441) for PLA2G7 identified in both platforms has a MAF of 0.06 in East Asians but <0.001 in other populations (Supplementary Fig. 15; Supplementary Data 4)[18]. Additionally, we identified *cis*-pQTLs for PCSK9 from Olink (sentinel variant: rs572512) and SomaScan (sentinel variant: rs151193009) that were not found in the European population, and which did not colocalise in CKB (Supplementary Fig. 16, Supplementary Data 4). The allele frequencies of those *cis*-pQTLs also vary greatly across populations, with the T allele frequency of rs572512 = 0.74 in East Asians and 0.34 in Europeans, and the T allele frequency of rs151193009 = 0.01 in East Asians and <0.001 in Europeans[18].

## Associations with adiposity and other traits

Overall BMI was significantly associated at FDR < 0.05 with 1096 (64.7%) Olink and 1429 (84.4%) SomaScan proteins (Fig. 3a; Supplementary Fig. 17). Applying Bonferroni correction reduced the number of significant proteins proportionally across different platforms (Supplementary Fig. 18). There was a moderate correlation of effect sizes (beta coefficients) for BMI between the platforms (Pearson's r = 0.51; 95%CI: 0.47–0.54) (Fig. 3b; Supplementary Fig. 19).

Of the 974 BMI-associated proteins in both Olink and SomaScan, 859 (88.2%) showed directionally consistent associations between the platforms (i.e. shared associations). Among these 859 proteins with shared associations, the between-platform correlation of BMI effect

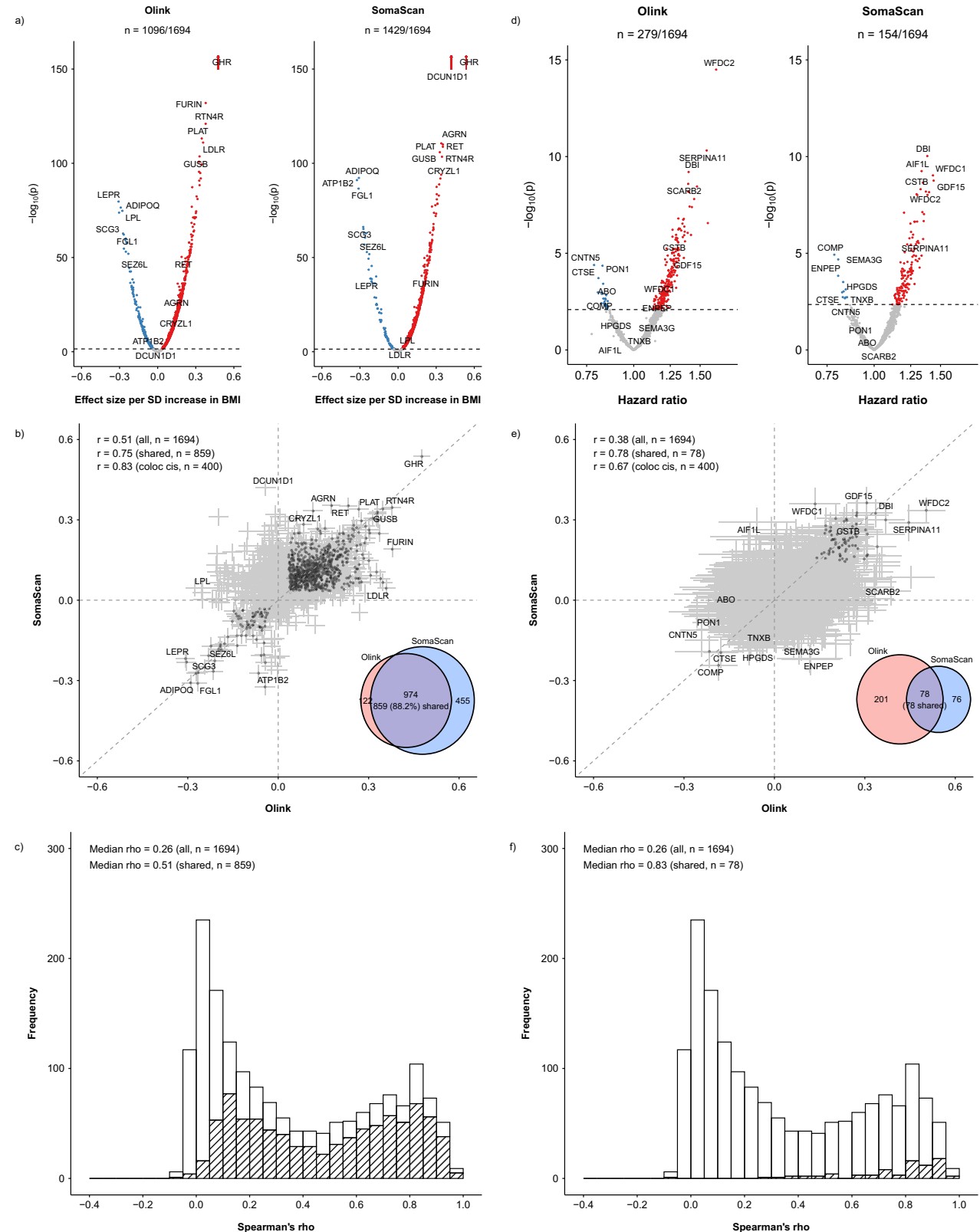

sizes was high (r = 0.75; 0.72–0.78; Fig. 3b), as was the median Spearman's correlation (rho = 0.51) across platforms (Fig. 3c; Supplementary Fig. 20). Similarly, among the 400 proteins with colocalising *cis*-pQTLs between Olink and SomaScan, there was a high between-platform correlation of effect sizes for BMI (r = 0.83; 0.80–0.86), with 251 (62.8%) having shared associations with BMI between the platforms. Further analyses revealed that proteins with lower observational

correlations, proteins of lower abundance, and proteins with more samples below LOD or with QC warning in Olink were more likely to have platform-specific associations or associations that had different directions of effect sizes between platforms. Interestingly, proteins with PAV *cis*-pQTLs were less likely to yield SomaScan-specific BMI associations compared to associations shared by both platforms (Supplementary Fig. 21).

**Fig. 3 | Comparison of number of proteins significantly associated with BMI and risk of incident IHD and their effect sizes between Olink and SomaScan platforms.** Analyses were conducted for 1694 matched proteins among 3976 participants, including 1951 IHD cases and 2025 subcohort participants. **a** Associations between protein levels and BMI, with coloured dots indicating significant associations. **b** Comparison of effect sizes (beta coefficients) for associations with BMI, with darker dots indicating significant associations and error bars indicating 95% confidence intervals. **c** Observational correlations between Olink and SomaScan, with shaded areas indicating shared associations for BMI. **d** Associations between protein levels and IHD. **e** Comparison of effect sizes (beta coefficients) for associations with IHD, with darker dots indicating significant associations and error bars indicating 95% confidence intervals. **f** Observational correlations between Olink and

SomaScan, with shaded areas indicating shared associations for IHD. Linear regression was used to test associations between protein levels and BMI, adjusted for age, age2, sex, ambient temperature (at sample collection) and its square, time since last meal and its square, plate ID, and case-subcohort ascertainment. Cox regression was used to test associations between protein levels and IHD, stratified by sex and region and adjusted for age, $age^2$, fasting time and its square, ambient temperature and its square, plate ID, education, smoking, alcohol consumption, physical activity, systolic blood pressure, type 2 diabetes, ApoB/ApoA, and BMI. $P$ values were corrected for multiple testing with a false discovery rate = 0.05 (two-sided) within each platform to define significant associations. BMI body mass index, IHD ischaemic heart disease.

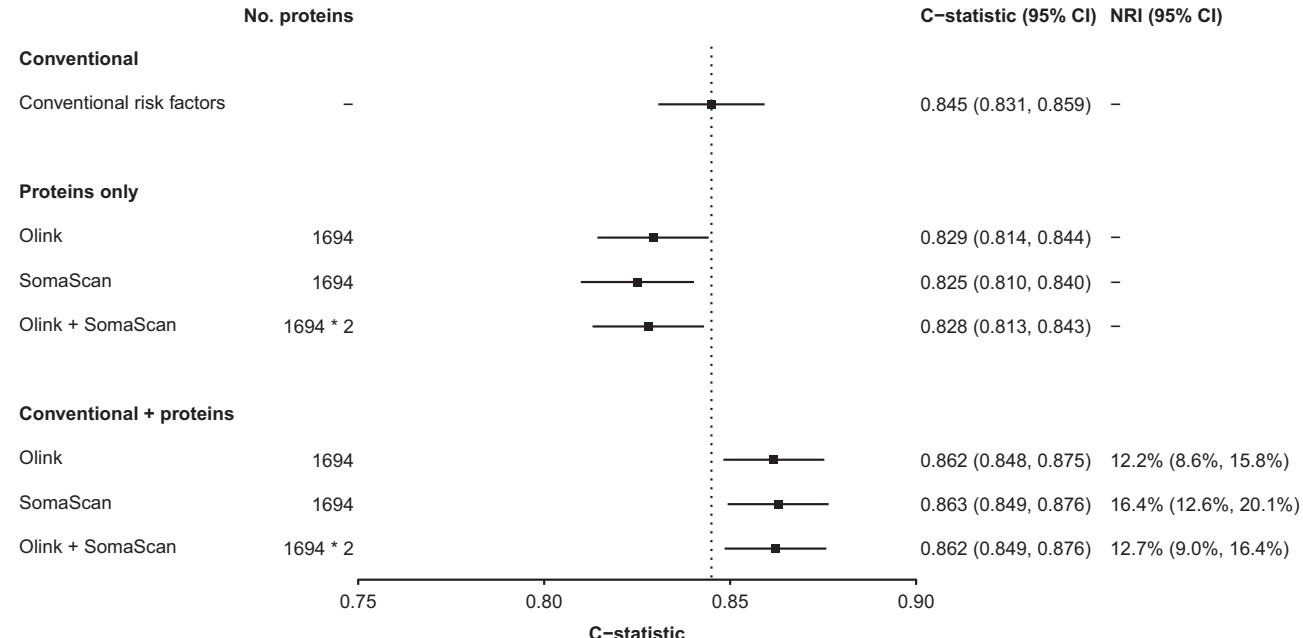

**Fig. 4 | Performance of proteins measured using Olink and SomaScan platforms for prediction of incident IHD.** Analyses were conducted for 1694 matched proteins among 3976 participants, including 1951 IHD cases and 2025 subcohort participants. Conventional risk factors for cardiovascular disease included age, sex, smoking, type 2 diabetes, systolic blood pressure, and waist circumference. Protein measurements for 1694 one-to-one matched proteins from both platforms or from one platform were used to build the models. Data are presented as estimated C-statistics, with error bars indicating 95% confidence intervals. NRI were computed with a decile-based method. IHD ischaemic heart disease, NRI net reclassification index.

Additional analyses highlighted the relevance of controlling for confounding by ambient temperature and time since last meal on Olink and SomaScan protein levels (Supplementary Note 3; Supplementary Figs. 22–28). The associations of proteins with a range of other traits and in subcohort participants are also shown in Supplementary Figs. 23–28 and Supplementary Data 5.

**Associations with incident IHD**
Overall, 279 (16.4%) Olink and 154 (9.1%) SomaScan proteins were significantly associated at FDR < 0.05 with risk of incident IHD (Fig. 3d; Supplementary Fig. 29; Supplementary Data 6). There were fewer significant associations with IHD after applying Bonferroni correction (Supplementary Fig. 30). All the 78 proteins that were significantly associated with IHD in both Olink and SomaScan also showed directionally concordant (shared) associations. For all overlapping proteins ($n = 1694$), the between-platform correlation (Pearson's r) of effect sizes was 0.38 (0.34–0.42), which increased to 0.78 (0.68–0.86) when restricted to proteins with shared associations with IHD (Fig. 3e; Supplementary Fig. 31). Proteins with shared associations with IHD also had higher between-platform Spearman's correlations (median rho = 0.83) (Fig. 3f; Supplementary Fig. 32). When the analyses were restricted to 400 proteins with colocalising *cis*-pQTLs, we found a

strong correlation of their effect sizes (r = 0.67; 0.62–0.72) with 45 (11.3%) proteins having shared associations with IHD between the platforms. Similar to the findings on BMI, proteins with lower observational correlations, proteins of lower abundance, and proteins with more samples below LOD or with QC warnings in Olink were also more likely to have platform-specific associations with IHD (Supplementary Fig. 33).

For the prediction of incident IHD, a risk prediction model based on all 1694 one-to-one matched proteins yielded similar performance between Olink (C-statistics: 0.829; 95%CI: 0.814–0.844) and SomaScan (C-statistics: 0.825; 95% CI: 0.810–0.840), which was slightly lower than a model that only included conventional risk factors (C-statistics: 0.845; 95% CI: 0.831–0.859). (Fig. 4; Supplementary Fig. 34). The addition of proteins to the conventional model increased the C-statistics to 0.862 (95% CI: 0.848–0.875) for Olink and 0.863 (95% CI: 0.849–0.876) for SomaScan. The net reclassification index (NRI) estimated using a decile-based method was 12.2% (95% CI: 8.6–15.8%) for Olink and 16.4% (95% CI: 12.6–20.1%) for SomaScan. Moreover, using all proteins from both platforms together did not significantly improve the model performance compared with using proteins from each platform separately (Fig. 4; Supplementary Fig. 34). Models using FDR-significant and significant proteins shared by both platforms showed

similar results as the model using all overlapping proteins (Supplementary Fig. 34). Additional analyses using a category-free method to estimate the NRI also yielded similar results as the decile-based method (Supplementary Fig. 35).

### Non-one-to-one matched reagent pairs

Of the remaining 474 proteins with non-one-to-one matched reagents, 472 were targeted by one Olink reagent and by 2–9 SOMAmers (1037 unique SOMAmers in total), resulting in 1053 Olink-SomaScan reagent pairs. We found a moderate correlation between protein levels measured by different SOMAmers targeting the same protein (median rho = 0.46; Supplementary Fig. 36). Likewise, there was a moderate correlation between those Olink and SomaScan reagent pairs (median rho = 0.37; Supplementary Fig. 37). Of those 472 proteins, 300 (63.6%) had at least one reagent pair with a Spearman's rho > 0.4. In GWAS, cis-pQTLs were identified for 277 Olink reagents and 399 SOMAmers, and trans-pQTLs were identified for 250 and 509 reagents, respectively. We found evidence for colocalisation of cis-pQTLs for 204 proteins (330 reagent pairs) between Olink reagents and varying numbers of their matched SOMAmers (Supplementary Tables 3, 4), while 192 proteins (308 reagent pairs) had colocalising trans-pQTLs between Olink reagents and their matched SOMAmers (Supplementary Data 3).

There were also 18 SOMAmers with each targeting two different proteins, resulting in 36 Olink-SomaScan reagent pairs. There were moderate correlations between protein levels measured by Olink reagents that were targeted by the same SOMAmer (median rho = 0.31; Supplementary Fig. 38) and between those Olink and SomaScan reagent pairs (median rho = 0.16; Supplementary Fig. 39). In GWAS, cis-pQTLs were identified by 9 Olink reagents and 3 SOMAmers, while trans-pQTLs were identified by 13 and 9 reagents, respectively. Only one of the 36 Olink-SomaScan reagent pairs showed evidence for colocalisation of their cis-pQTLs, and three pairs (from three different proteins) showed evidence for colocalisation of their trans-pQTLs (Supplementary Data 3).

## Discussion

In this large-scale direct comparative analysis of 2168 proteins measured by both Olink and SomaScan platforms in Chinese adults, we found only modest correlations between plasma levels of proteins measured by each platform. In GWAS, Olink proteins were more likely to have cis-pQTLs compared with SomaScan proteins, and >75% of the proteins with cis-pQTLs in both platforms had signals which colocalised. SomaScan yielded more adiposity-associated proteins than Olink, but the majority of these were shared across platforms. In contrast, Olink had more IHD-associated proteins than SomaScan, but their overall performance for risk prediction of IHD was similar. We also identified a group of proteins that showed high correlations and consistent genetic and phenotypic associations between Olink and SomaScan, and their replicability across platforms may assist with interpretation of future study findings.

Previous studies in Western populations designed to compare Olink and SomaScan platforms had varying numbers of participants (10–1514) and numbers of overlapping proteins (425–1848)[13–17]. Most studies reported only moderate correlations (median rho ≈ 0.4) of protein levels between each platform[13–16]. Two previous studies examined the concordance of pQTLs identified by the two platforms in the same sample. One study showed half of the proteins with cis-pQTLs had signals discovered in both platforms[16], while another found that 64% of genomic region-protein associations were shared based on LD[15]. More recently, Eldjarn et al. reported that about half of the cis-pQTL signals in one platform had a corresponding pQTL in the other, although their pQTLs were identified not in the same study population but in two different datasets[17]. Unlike the present study, previous studies did not analyse proteins separately that were targeted by multiple reagents, which showed different patterns of associations,

nor assessed the consistency of pQTLs using colocalisation analyses. In the present study, we found more modest correlations (median rho = 0.29) between the two platforms than previous studies, possibly reflecting the inclusion of a higher proportion of low-abundance proteins in our study. Indeed, in analyses stratified by protein abundance, there were stronger correlations (median rho ≈ 0.6) for more highly abundant proteins (i.e. those with dilution factor < 10%; Supplementary Figs. 8, 9). In our GWAS, Olink identified slightly more proteins with cis-pQTLs than SomaScan. This could be due to differences in binding specificity between Olink and SomaScan reagents and the protein isoforms they detect. In the colocalisation analyses, we found that about 50% of the proteins with cis-pQTLs obtained in at least one platform colocalised between Olink and SomaScan, with the proportion increasing to over 75% when analyses were restricted to those with cis-pQTLs discovered in both platforms. We also identified a large number of proteins with trans-pQTLs in both platforms. Analyses of trans-pQTLs could identify important regulatory pathways for such proteins, but the possibility of non-specific binding of reagents or non-specific pleiotropic variants cannot be excluded.

The present analyses conducted in an East Asian population identified pQTLs that had not been previously reported in other ancestry populations. Indeed, our previous analysis of the Olink proteins in the same study population indicated that >30% of the identified credible sets of pQTLs did not overlap with those found in participants of European ancestry in the UK Biobank[19]. Although technical differences between studies might explain such differences in pQTLs, since CKB and the UK biobank used the same Olink platform which showed high reproducibility[12,13], it is likely that these ancestry-specific pQTLs can be mainly attributed to the different genetic architecture between East-Asian and European populations. Ancestry-specific pQTLs can be particularly informative when used as genetic instruments in Mendelian Randomisation (MR) studies for causal inference and drug target discovery in diverse populations. For example, variation in the ALDH2 gene is known to have a significant impact on alcohol metabolism and thus relates to the East Asian-specific flushing syndrome[20]. Previous reports from the CKB study found that genetically predicted alcohol intake based on the cis-pQTL for ALDH2 was causally associated with multiple diseases and cause-specific mortality in Chinese men[20–22]. Furthermore, although previous observational studies suggested a role for Lp-PLA2 (PLA2G7) in the aetiology of cardiovascular diseases, inflammation, and life span[23,24], the PLA2G7 cis-pQTL was unrelated to risks of major vascular or non-vascular diseases in Chinese adults in CKB[25], consistent with the findings of clinical trials of LpPLA2-lowering medication[26]. It is worth noting that cis-pQTLs for ALDH2 and PLA2G7 are both PAVs, and could be related to potential epitope effects (i.e. genetic variants influencing protein structure and hence reagent binding, instead of protein levels). However, it's unlikely that these two cis-pQTLs are explained by epitope effects alone, as previous studies have reported the effects of these variants on the concentrations of ALDH2 and PLA2G7 proteins measured by different immunoassays[27–29], and our previous studies in CKB also demonstrate a functional impact of the cis-pQTL of ALDH2[20–22]. Nevertheless, extra caution should be taken when cis-pQTLs are used as genetic instruments in MR studies, as they might only influence assay reagent binding instead of actual plasma levels of the corresponding protein.

In addition to ancestry-specific cis-pQTLs that colocalised between Olink and SomaScan, we also identified ancestry-specific cis-pQTLs that did not colocalise in CKB. For example, although we identified cis-pQTLs for PCSK9 in both platforms, these did not colocalise. Moreover, neither of them was found in previous studies in European populations, in which the cis-pQTLs for PCSK9 were replicated between Olink and SomaScan[9,12,15–17]. Such differences in platform concordance might also be explained by differences in genetic architecture across populations: the allele frequency of the CKB Olink

*cis*-pQTL sentinel variant for PCSK9 varies between East Asian and European populations, while the CKB SomaScan *cis*-pQTL sentinel variant for PCSK9 is very rare in European populations. Since the latter variant is a missense variant, it might reflect ancestry-specific epitope effects, although we could not directly test this hypothesis without more information on reagent binding.

Few previous studies have systematically compared the associations of Olink and SomaScan proteins with traits and risks of disease outcomes in the same sample. In the present study, SomaScan yielded more BMI-associated proteins compared with Olink, while for IHD risk, the converse was true. Importantly, the number of BMI-associated proteins (and several other baseline characteristics) was attenuated when the analyses were conducted using SomaScan-ANML data, which normalises protein levels within the 'normal' range based on an external reference (healthy individuals) while preserving outliers (which may be affected by disease status)[7,8]. This is in contrast with the pQTL analyses, where SomaScan-ANML identified more pQTLs than SomaScan-non-ANML, as revealed by the current study and previous analyses by deCODE[17]. We presented the SomaScan-non-ANML data as the main results, as they were more comparable to Olink data than the SomaScan-ANML data. Nevertheless, regardless of the version of the SomaScan data used, we found that proteins with high correlations between the platforms or with colocalising *cis*-pQTLs tended to yield replicable associations with traits and disease risk across platforms, as shown in Supplementary Fig. 40 and Supplementary Data 7. This can be illustrated by *GHR* (associated with BMI in both platforms) and *WFDC2* (associated with IHD in both platforms). Both proteins showed high concordance in their observational correlations and *cis*-pQTLs (Supplementary Data 4), and their associations with adiposity or with cardiovascular diseases have been previously reported[30–32]. This consistency across different platforms will inform researchers when interpreting findings on these proteins, which may be more likely to reveal true biological signals.

The present study was the first study to directly compare the performance of Olink and SomaScan proteins for risk prediction of major diseases. Although proteomic prediction models alone did not outperform the performance of the model with conventional risk factors for IHD in analyses of a set of proteins captured by both platforms, the addition of proteins to conventional risk factors improved risk prediction of IHD. Adding Olink and SomaScan proteins to conventional risk factors showed comparable performance with similar improvement in C-statistics and positive NRIs. Moreover, the use of all overlapping proteins from both platforms together did not significantly improve the model performance, suggesting that using a set of core proteins from one platform alone may be suitable for risk prediction, at least for IHD. Nevertheless, the present analyses focused only on the 1694 overlapping proteins, but there are more proteins measured by the Olink and SomaScan assays that are platform-sepcific, which may further improve risk prediction[33,34].

Consistent with previous studies[15], our Boruta feature selection analyses demonstrated that protein abundance and data quality (e.g. % below LOD and QC warning) were the two most important factors that accounted for the concordance between measurements, as well as differences in downstream genetic and observational analyses between Olink and SomaScan platforms. This was further confirmed by our analyses showing that proteins with lower observational correlations, proteins of lower abundance, and proteins with more samples below LOD or more samples with QC warnings in Olink were more likely to yield platform-specific or distinct findings between platforms. In particular, while most (>85%) proteins that were significantly associated with BMI or IHD in both platforms showed directionally consistent effects with a high (r > 0.7) correlation of effect sizes, some BMI-associated proteins showed discordant directions of effects between the two platforms. The latter proteins typically had low correlations (median Spearman's rho = 0.04), possibly reflecting poor data quality.

Conversely, protein characteristics based on annotations from UniProt Knowledgebase and Gene Ontology (GO) contributed little to the differences between the platforms, at least for the 1694 proteins with one-to-one matched single reagents.

Nevertheless, discordant findings between platforms might have been influenced by reagents binding to different proteoforms of the same protein, leading to epitope effects and differences in pQTLs[9,17]. Indeed, our additional analyses showed that compared to SomaScan *cis*-pQTLs that were not related to PAVs, SomaScan PAV *cis*-pQTLs were more likely to not colocalise with the Olink *cis*-pQTLs identified for the same protein, suggesting potential epitope effects (Supplementary Fig. 13). Further analyses of 474 proteins matched to more than one SomaScan or Olink reagent could also explore possible epitope effects. For example, not all SOMAmers targeting TNC showed evidence for colocalisation of *cis*-pQTLs with the corresponding Olink reagent (Supplementary Figs. 41, 42; Supplementary Data 4). This could be a result of those reagents targeting different structures of TNC, a protein known to have different isoforms, while several *cis*-pQTLs identified for TNC in SomaScan were indeed in LD with PAVs (Supplementary Data 3, 4)[35,36]. However, in the absence of any data on proteoforms for either platform, it was not possible to reliably assess this hypothesis. Finally, we found that proteins with PAV *cis*-pQTLs were more likely to yield BMI associations shared by both platforms compared to SomaScan-specific associations, (Supplementary Fig. 21). This could be because PAV *cis*-pQTLs, despite potential epitope effects, might also have more functional impact on phenotypes, which could be reliably detected by both platforms. More research is needed to further explore the influence of PAVs on pQTLs and phenotypic associations.

The present study is the largest study to date that directly compared the utility of antibody and aptamer-based proteomic platforms in identical blood samples. We assessed the agreement between the platforms using both observational and genetic analyses, and compared analytical performance for their associations with traits and IHD, in addition to utility for risk prediction of IHD. However, the present study also had several limitations. First, we did not compare coefficients of variation (CV) as a measure of accuracy, as these are strongly influenced by the data distribution. Moreover, we were unable to adopt the CV ratio (i.e. CV in repeated samples divided by CV in unrelated samples) as proposed by Eldjarn et al. to compare accuracy between the platforms[17], due to the small number of duplicates of proteins/samples measured in the study. Second, the analyses chiefly focused on one-to-one matched Olink-SomaScan reagent pairs to allow for a straightforward comparison, but reagents from different platforms could still target different proteoforms of the same unique protein. Further analyses of proteins assayed using different reagents targeting their different proteoforms may offer additional insights, particularly those related to possible epitope effects[15]. Third, we only compared 2168 overlapping proteins matched to the same UniProt ID based on the previous version of each platform. Recently, the number of proteins measured by Olink has increased to over 5000, while the latest SomaScan platform includes about 11,000 SOMAmers, and this is likely to increase the number of overlapping proteins between the platforms. Fourth, previous studies showed that pQTLs which are PAVs but are not eQTLs could indicate epitope effects[15,17]. However, we were unable to directly assess this due to the lack of available eQTL studies in the East Asian population. Finally, we only compared two major affinity-based platforms. Future studies will be required to compare agreement with measurements from mass spectrometry platforms, which remains the 'gold standard' for protein identification. With advances in mass spectrometry that increase assay throughput[37,38], future comparative studies of mass spectrometry with affinity-based platforms are now warranted.

Overall, this study assessed the analytical performance of two affinity-based proteomic platforms in a large study of Chinese adults. As each platform has its strengths, the selection of platforms in future

studies may depend on the study purpose (e.g. mechanistic investigation vs risk prediction), analytical approach (e.g. observational vs genetic), preferred breadth of coverage of the platform, and overall cost-effectiveness. For future studies employing only one platform, the findings on proteins with high replicability in the current study might be interpreted more confidently, given their consistency across platforms. Since affinity-based and mass spectrometry technologies are still evolving and will likely capture more overlapping proteins between different platforms, further research is still needed to compare, both directly and indirectly, the utilities of different platforms in order to inform large-scale population and clinical research.

## Methods

### Study population and design

The CKB is a prospective cohort study with >512,000 adults recruited during 2004–08 from 10 geographically diverse areas[39]. At baseline and subsequent periodic resurveys of a random subset of participants, detailed data were collected from participants using laptop-based questionnaires (e.g. socio-demographic characteristics, medical history, and lifestyle habits) and physical measurements (e.g. anthropometry, blood pressure, heart rate, lung function). Non-fasting (with time since the last meal recorded) blood samples were collected, processed, aliquoted, and then stored in liquid nitrogen. After the initial baseline survey, the long-term health of the participants was monitored by linkage with local death or disease registries and with the national health insurance systems that record any episodes of hospitalisation[39]. The CKB complies with all required ethical standards for medical research on human subjects. Ethical approval was obtained from the Ethical Review Committee of the Chinese Centre for Disease Control and Prevention (Beijing, China, 005/2004) and the Oxford Tropical Research Ethics Committee, University of Oxford (UK, 025-04). All participants provided written informed consent.

The present analysis involved a case-subcohort study of 3977 unrelated participants (1951 incident IHD cases and 2026 subcohort participants) who were genotyped and had no prior history of cardiovascular diseases[40]. One subcohort participant was excluded due to insufficient sample volume for SomaScan.

### Genotyping

A total of 100,706 CKB participants were genotyped using a custom Affymetrix array, with 531,565 variants passing QC. They were converted to genome build 38 using CrossMap v0.6.1[41] and checked for consistency by reversing the process (liftUnder). Variants were excluded if they were not mapped, mapped to different chromosomes, or not mapped back to the same locations after liftUnder, leaving 531,542 remaining variants for further analyses. They were pre-phased using SHAPEIT v4.2 (SHAPEIT v2.904 for chromosome X)[42] and uploaded to the TOPMed[43] or Westlake Biobank for Chinese[44] server for imputation. Two sets of the imputed data were merged by selecting the imputed genotype with a higher imputation INFO score for each variant. Variants with an INFO score < 0.3 or MAF = 0 were excluded. Details of the genotyping and QC procedures have been previously described[45].

### Proteomic assays

For the Olink Explore 3072 assay, stored baseline plasma samples for 3977 participants were retrieved from liquid nitrogen and thawed, and 40 μl plasma was aliquoted into 96-well plates (including 8 wells per plate for external QC samples). Plates were shipped in two batches for assay at Olink Laboratories first batch, 1472 proteins in Uppsala, Sweden; and second batch, 1469 proteins in Boston, USA. Protein levels were normalised based on inter-plate controls and transformed using a pre-determined correction factor. LOD for reagents was defined using external QC samples. QC warnings for participant samples and assay warnings for plates were flagged based on deviations in the QC

samples[46]. Protein levels were provided in the arbitrary Normalized Protein eXpression (NPX) unit on a log2 scale. Among a total of 2941 protein reagents, six were replicated across all four panels, resulting in 2923 unique reagents. Only one measure for each duplicated Olink reagent was used for the comparative analysis, as replicated protein levels had high correlations between panels (r > 0.8).

For the SomaScan v4.1 assay, 60 μl plasma aliquots in 2D-barcoded microtubes for the same 3977 participants were sent to the Somalogic Laboratory in Colorado, USA for profiling by SomaScan Assay v4.1, which covers a total of 7596 SOMAmers, including 7301 with available UniProt IDs. Samples were randomised at the Somalogic laboratory and aliquoted into 96-well plates (11 wells allocated for external control samples, including 5 calibrator, 3 QC, and 3 buffer samples). One subcohort participant was excluded due to insufficient sample volume. Three serial dilutions (0.05%, 0.5%, and 20%) were conducted to achieve optimal detection of protein targets, and each SOMAmer was only present in one of the three dilutions, which was consistent across all participants. For 91 participants with higher sample volumes, samples were split and run in duplicate. Only one measure from each duplicated sample was used for the comparative analyses as protein levels between duplicate samples were highly correlated (median rho > 0.8). The raw SomaScan assay results were standardised based on external control samples to control for variability in microarrays and variation within and across plates. This also included an optional step of adaptive normalization by maximum likelihood (ANML) to an external reference to control for inter-sample variability[47]. The final SomaScan data were supplied in both ANML and non-ANML versions in relative fluorescence units (RFU), which were further log-transformed (natural log) in the main analyses. The LOD for SOMAmers was defined using external buffer samples. QC checks were performed by comparing the median of QC samples on each plate to the reference, and a cross-plate QC check measure (pass/flag) was assigned to each SOMAmer.

### Protein target mapping and reagent matching

We mapped Olink reagents and SOMAmers to proteins based on their UniProt[48] IDs provided by Olink and SomaScan[48]. If two reagents from different platforms were mapped to the same UniProt ID, we considered them to be an Olink-SomaScan reagent pair. The main analysis focused on reagent pairs where one unique Olink reagent was matched to one unique SOMAmer. Separate analyses were also undertaken to investigate proteins for which single SOMAmers matched to multiple Olink reagents and single Olink reagents matched to multiple SOMAmers.

### Statistical analyses

We calculated Spearman's and Pearson's correlation coefficients for each Olink-SomaScan reagent pair. We annotated all Olink-SomaScan reagent pairs in four areas, including: (i) Olink or SomaScan assay-related factors (e.g. batch, dilution factor); (ii) Olink or SomaScan data-related measures (e.g. % outliers [>4 SD from the mean], % below LOD); (iii) protein characteristics retrieved from the UniProt Knowledgebase[48] (e.g. presence of a transmembrane domain); and (iv) GO annotations[49]. Following the method used in Pietzner et al.[15], we employed Boruta feature selection[50], a random-forest-based machine learning approach, to identify factors that were predictive of Spearman's rho. We included a total of 87 features in Boruta (for GO, the top 10 most annotated terms from each GO category), with a p-value threshold of 0.01 and a maximal number of 50,000 runs. An importance measure was generated for each variable after each run. For features confirmed by Boruta, we further tested their associations with Spearman's rho using linear regression to determine the direction of their effect.

We performed GWAS of each protein, with genotyping array version, age, age², sex, region, and ten national genomic principal components included as covariates. Protein levels were analysed as

rank inverse normal transformed residuals following linear regression on sex, age, age2, and region. GWAS were conducted using BOLT-LMM v2.3.4[51] (REGENIE v3.2.5.2[52] when BOLT-LMM failed due to low heritability) for Olink and REGENIE v3.2.5.2[52] for SomaScan. Only SNPs with INFO > 0.3 and minor allele count (MAC) > 20 were included. Associated loci were defined by genome-wide significant variants ($p < 5 \times 10^{-8}$) after linkage disequilibrium (LD) clumping using PLINK[53,54] (initial window ±10 Mbp [40 Mbp for the MHC region], $p < 0.05$, $r^2 > 0.05$), based on an internal LD reference of 72,000 unrelated CKB participants. Overlapping loci were merged and extended by ±10 kb, and the variant with the lowest $p$-value was identified as the sentinel variant. pQTLs within 500 kb on either side of the protein-encoding gene were defined as *cis*-pQTLs, while pQTLs outside this window were defined as *trans*-pQTLs. For *cis*-pQTLs discovered in at least one platform, we performed colocalisation using coloc (v5.2.1)[55] with the coloc.abf function under a single causal variant assumption, based on summary statistics of Olink and SomaScan proteins. A PPH4 > 0.8 was considered to be evidence of colocalisation. We also estimated the $r^2$ between sentinel *cis*-pQTLs for proteins with overlapping *cis*-pQTLs identified in both platforms. Based on Ensembl Variant Effect Predictor (VEP)[56], we used BCFtools[57] to annotate the sentinel variants of all identified *cis*-pQTLs, as well as their proxies with $r^2 > 0.8$ and within a 500 kb window on either side. Variants were defined as PAVs if they fell within one of the following categories: splice acceptor, splice donor, stop gained, stop lost, frameshift, start lost, variant, inframe insertion, inframe deletion, and protein altering.

We conducted linear regression analyses to examine cross-sectional associations between protein levels and selected baseline traits, with scaled protein levels (i.e. divided by SD) as the outcome and traits of interest (e.g. BMI, heart rate, smoking) as explanatory variables. The models were adjusted for age, age[2], sex, region, ambient temperature (at sample collection) and its square, time since last meal and its square, plate ID, and case-subcohort ascertainment. We employed the Benjamini–Hochberg method to control the false discovery rate (FDR) in multiple testing within each trait and each platform[58]. The main analyses included all participants, with additional sensitivity analyses restricted to the subcohort participants only. Analyses of associations of proteins with ever regular smoking and regular alcohol drinking were limited to male participants (as very few females in CKB smoked or drank alcohol). Participants with missing data were excluded from the corresponding analysis.

We used the Prentice pseudo-partial likelihood Cox regression method for analyses of associations of levels of proteins with incident IHD (ICD-10 codes: I20, I22–I25)[59]. All models were adjusted for age, age[2], sex, region, ambient temperature and its square, time since last meal and its square, plate ID, education, physical activity, alcohol intake, smoking, systolic blood pressure, diabetes, and BMI. Subcohort participants were censored at the time of any other IHD diagnosis[60].

For risk prediction of IHD, we used LASSO logistic regression models using four sets of proteins: (i) all matched proteins from both platforms; (ii) all matched proteins from one platform; (iii) significant proteins in each platform after FDR correction; (iv) significant proteins that were shared between the platforms. For risk prediction, plasma proteins levels were centred by plate using the median of sub-cohort values, and missing values were mean imputed. For estimation of discrimination, five repeats of 10-fold cross-validation were used (in which the study sample was split into 90% training and 10% test datasets, stratified by IHD status at the end of follow-up and ascertainment). In this way, the training and test datasets were always independent of each other each time. Within each training data set, the lambda (regularisation) hyperparameter was selected by applying the one standard error rule to the AUC (i.e. largest lambda value such that AUC is within 1 standard error of the minimum) using 10-fold cross-validation. This ensured that hyperparameter optimisation was also independent of the test dataset. Region-specific intercept terms were used in all models, with no penalty applied to these terms. Missing values were mean imputed within training data sets. Discrimination performance was measured by Harrell's concordance index (C-statistic) (including appropriate subject pairs as per the case-cohort design) stratified by region[61]. Reclassification was measured using both decile-based[62] and category-free[63] net reclassification index (NRI) compared to a prediction model including conventional risk factors previously developed in CKB (age, sex, smoking, type 2 diabetes, systolic blood pressure, and waist circumference)[64]. Models were trained on the overall sample on 25 separate iterations (using different splits for the cross-validated selection of the regularization parameter) and the median NRI was reported for each model comparison.

To identify factors that might explain the discordant findings between Olink and SomaScan, we firstly categorised proteins into four groups based on whether they had Olink-specific, SomaScan-specific, distinct, or shared *cis*-pQTLs, BMI associations, or IHD associations. We then used logistic regression to test if observational correlations, key technical factors, or PAVs could explain any of the observed discordant findings, using proteins with shared *cis*-pQTLs or associations as the reference group. Analyses on PAV *cis*-pQTLs were restricted to proteins with *cis*-pQTLs identified in the corresponding platform.

All statistical analyses were conducted in R (v4.2.2)[65] and their significance was determined using two-tailed tests[66].

### Reporting summary
Further information on research design is available in the Nature Portfolio Reporting Summary linked to this article.

## Data availability
In CKB, non-genetic data (e.g. baseline, resurveys, biomarkers, and disease endpoints) are available and updated periodically for access by bona fide researchers. Details of the CKB Data Sharing Policy, data release schedules and data request application procedures are available at www.ckbiobank.org. All queries about data access can be made to ckbaccess@ndph.ox.ac.uk. Accessing to individual participant genetic data (e.g. genotyping, whole genome sequence) is currently constrained by China's Administrative Regulations on Human Genetic Resources, for which collaboration with CKB researchers is generally required, which may be subject to separate regulatory approvals in China if it involves substantial sharing of unpublished data. The proteomic data analysed in this study are available at https://doi.org/10.6084/m9.figshare.27931350. The pQTL summary statistics of the proteins analysed are provided in Supplementary Data 4. The full pQTL summary statistics, together with other CKB GWAS summary statistics, can also be accessed through the CKB PheWeb browser (https://pheweb.ckbiobank.org/). The fully linked and integrated proteomic data with other data will be made available through the CKB Data Access System during 2025. Further information is available from the corresponding authors upon request. All data generated in this study are provided in the Supplementary Data files. Researchers who are interested in obtaining the raw individual participant data (including proteomics data) related to this paper can contact ckbaccess@ndph.ox.ac.uk. Further information is available from the corresponding authors upon request.

## Code availability
Genome-wide association analyses were performed using publicly available software packages (BOLT-LMM v2.3.4: https://alkesgroup.broadinstitute.org/BOLT-LMM/BOLT-LMM_manual.html; regenie v3.2.5.2: https://rgcgithub.github.io/regenie/). Comparative analyses were conducted using R (v4.2.2; https://www.r-project.org/) and custom code can be accessed through https://github.com/baihanwang211/proteomics-public.

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

## Acknowledgements

The chief acknowledgement is to the participants, the project staff, and the China CDC and its regional offices for assisting with the fieldwork. We thank Judith Mackay in Hong Kong; Yu Wang, Gonghuan Yang, Zhengfu Qiang, Lin Feng, Maigeng Zhou, Wenhua Zhao, and Yan Zhang in China CDC; Lingzhi Kong, Xiucheng Yu, and Kun Li in the Chinese Ministry of Health for assisting with conduct and organization of the study. The CKB baseline survey and the first re-survey were supported by the Kadoorie Charitable Foundation in Hong Kong. The long-term follow-up and subsequent resurveys have been supported by Wellcome grants to Oxford University (212946/Z/18/Z, 202922/Z/16/Z, 104085/Z/14/Z, 088158/Z/09/Z) and grants from the National Natural Science Foundation of China (82192901, 82192904, 82192900) and from the National Key Research and Development Programme of China (2016YFC0900500). The UK Medical Research Council (MC_UU_00017/1, MC_UU_12026/2, MC_U137686851), Cancer Research UK (C16077/A29186, C500/A16896) and British Heart Foundation (CH/1996001/9454), provide core funding to the Clinical Trial Service Unit and Epidemiological Studies Unit, Oxford University for the project. The proteomic assays were supported by BHF (FS/18/23/33512), Novo Nordisk, Olink, SomaScan and NDPH. DNA extraction and genotyping were supported by GlaxoSmithKline and the UK Medical Research Council (MC-PC-13049, MC-PC-14135). Computation used the Oxford Biomedical Research Computing (BMRC) facility, a joint development between the Wellcome Centre for Human Genetics and the Big Data Institute supported by Health Data Research UK and the NIHR Oxford Biomedical Research Centre. The views expressed are those of the author(s) and not necessarily those of the NHS, the NIHR or the Department of Health.

## Author contributions

B.W., R.W., R.C., I.M., and Z.C. contributed to the concept and design of this study. B.W., A.P., M.M., N.W. conducted the statistical analyses and B.W., I.M. and Z.C. drafted the manuscript. B.W., A.P., M.M., N.W., P.Y., S.S., A.I., C.K., H.F., K.L., Y.C., H.D., D.A., D.S.V., C.Y., D.S., J.L., M.H., L.L., D.B., R.C.o., R.W,. R.C., I.M., and Z.C. were involved in the planning, acquisition and interpretation of data. H.F., K.L., Y.C., H.D., D.A., and D.S. and provided administrative, technical, or material support. All authors provided critical revision of the manuscript for important intellectual content. B.W., A.P., M.M., N.W., I.M. and Z.C. are the guarantors of this work and take responsibility for the integrity and accuracy of the data analysis. I.M. and Z.C. supervised the work.

## Competing interests

The authors declare no competing interests.

## Additional information

## China Kadoorie Biobank Collaborative Group

Baihan Wang[1], Alfred Pozarickij[1], Mohsen Mazidi[1], Neil Wright [1], Pang Yao[1], Saredo Said[1], Andri Iona[1], Christiana Kartsonaki[1], Hannah Fry [1], Kuang Lin[1], Yiping Chen [1], Huaidong Du [1], Daniel Avery [1], Dan Schmidt-Valle[1], Canqing Yu [2,3,4], Dianjianyi Sun [2,3,4], Jun Lv[2,3,4], Liming Li [2,3,4], Derrick A. Bennett [1], Rory Collins[1], Robin G. Walters [1], Robert Clarke [1], Iona Y. Millwood [1,5] ✉ & Zhengming Chen [1,5] ✉

