## [Peer Review File · Nature Communications]

Comparative studies of 2,168 plasma proteins
measured by two affinity-based platforms in 4,000
Chinese adultsREVIEWER COMMENTS

Reviewer #1 (Remarks to the Author):

The authors of "Comparative studies of genetic and phenotypic
2 associations for 2,168 plasma proteins measured by two
3 affinity-based platforms in 4,000 Chinese adults" present a clear manuscript on important topics ;
proteophenomics and proteogenomics.

I have two main remarks :

A/ the first one is about the chronology of the presentation . The authors announce 2168 protein
matched by both assays but then go into 1694 proteins where both Olink and Soma are attempting a
single reagent each. Then discuss the close to 500 where there is not a one to one match.
I very strongly recommend that the authors present first an overall statistics of what is the main result
for the 2168. Then they can go and split it. One way to count each protein once and to wait by the
inverse of number of pairs. The data are currently presented in an artificial manner.
Lets remeber that even if protein names are unique, ther is likely multiple proteoforms per protein.
(isoform, bound vs free, cleaved , uncleaveed)
Related to that topic in the chapter, line 374, about the proteins with non one to one pairing they
should take the opportunity to go say how often there is at least one pair over lets say 0.4 and when
all are lower.

B/ the second comment is more about results and points not currently in the manuscript that I think
would gretaly improve the manuscript. First, let me talk about my first impression and the one of
readers when seeing a title on proteogenomics among Chinese: we all appreciate a greater ethnic
diversity in genomic studies currentlty dominated by studies among European subjects.

I missed to see an angle in the results about the Ethnic angle

Follwiwng that comment I have three suggestions and one extra point?

B1- I suggest that the authors attempt to compare LD class of pQTL in Chinese vs European. Is there
any ancestry refinement going one or the other direction. OD the authors have nice examples?

B2- There is a number of GWAS of diseases in East Asian including Chinese. Can the authors make an
attempt of colocalising pqtI(mainly cis) with disease association in the same ethnic group

B3-Do the authors can think of any diseases that is more common in Chinese and do the amse as in
B2

Currently the ethnic angle is limited to discussion. I suggest it to be main result section.

The following extra point is not a review poin, more a mention of my surprise:

Checking the affiliation of authors, I was a surprised to see that only four of the around 25 individually
listed authors where affiliated in China. I am aware of the CKB group last authorship.

Overall, this is a very good paper, that I think would greatly benefit from

- presenting the overall comparison and not focus in abtract on the 1694 but go straight to the 2168
proteins

- investigating and presenting resultis on the strength of the paper which is the population.

Reviewer #2 (Remarks to the Author):

This paper presents a head-on comparison of the two major affinity proteomics platforms, SOMAscan

and Olink, on a large number of samples of Chinese origin (N=4000). Comparison is performed by direct correlation and also by shared association with genetic and non-genetic traits. The paper addresses an important question as both platforms are increasingly used in large scale population studies, but previous smaller scale studies suggested that the agreement between both platforms is less than expected. Different reasons for these disagreements have been proposed, including differences in target binding of the affinity probes and detection limits. While this paper does not answer all of these questions, it advances our insights substantially.

Most striking is the finding of a bimodal distribution of Spearman's rho correlation presumably identical protein targets, with a peak centered around 0 and another closer to 0.8. This should be emphasized and discussed further, rather than stating in the abstract "there was a modest between platform correlation (median rho=0.20)". The correlation between the platforms is actually quite good, but only on a smaller subset of protein targets.

Taken together this is a very relevant paper that will eventually be of great interest to the wider readership of Nature Communications.

However, there are a number of points where the paper could be improved, both regarding clarity and comprehensiveness.

Specific points:

ANML versus non-ANML: To make the paper easier to read, use and discuss only one of the two normalization methods provided by SOMAscan (ANML or non-ANML) in the main text and move the comparison of ANML to non-ANML to the supplement (or remove it entirely). This is interesting but distracts from the main question. Removing one will also render the Figures more readable, as there will be less plots to show. Also decide from the start on log-scaling the SOMA data. There is no need to show that log-scaling works better, as the OLINK data is already on a log-scale.

Use Spearman correlations throughout for consistency (or Pearson, but not multiple methods).

Matching of proteins between the platforms: The authors use UniProt ids to identify orthologous targets and limit the major part of the study to proteins with a 1-1 match. However, I would also be interested in knowing which proteins actually are the best mutual hits on a pure correlation basis, assuming of course that 1-1 matches should also be their best mutual best correlation hit. Is that always the case? Are there exceptions? This would also allow to address the question as to which of the multiple SOMA targets are actually orthologs of the OLINK targets, which could then be added to the general analysis. By mutual best hit I understand pairs of proteins where one is the highest correlate of the other and vice versa (based on an all-against-all correlation between platforms). If a protein is only present on one platform, then its mutual best hit can of course be another protein that is related to the same pathway (which is also of interest). I believe a table of mutual best hit correlations would be of great interest. The concept is borrowed from the definition of orthologous genes between bacteria. I would encourage the authors to proceed to such an analysis, as it validates the protein match based on UniProtID. It may also identify binders that target the wrong protein, as happened on the SOMAscan platform in the past when some aptamers were developed on a wrongly labeled protein sample (also a possible reason for lack of correlation).

What is the rationale of using ambient temperature at sample collection as co-variate? How strong is that signal? The reason I am asking is that we saw a strong association with what Somalogic calls "cell abuse" in PCAs. One proxy for this kind of cell abuse is HSP90, and adding HSP90 as a covariate significantly improved many associations. The hypothesis is that some of the proteins originate from white blood cells, and then that white blood cells are more frequently spilling their content after collection at higher ambient temperatures. It would also be interesting to run a PCA on the proteins and investigate whether one of the major components correlate with HSP90, other proteins labeled as markers for "cell abuse" by Somalogic, or ambient temperature. More generally, it would be a good idea to correlate the first PCs from both platforms to check for shared general variance, be it technical or biological.

Please provide individual scatter plots for all proteins as a supplementary PDF. It would be important to show the raw data at some point without readers having to write a biobank application first (consider sharing the proteomics data outside a biobank application, if that is possible – final SOMA and OLINK datasets are small enough to be provided as Supplementary Tables).

This paper comprises a full pQTL study, but little data is actually shown. What are the plans for sharing the GWAS summary statistics? How do the pQTLs compare to published pQTL studies in Europeans? Further analysis of these might be the topic of another paper, but since the pQTLs are central to the comparison, this needs to be covered to a certain extent.

The authors state that "The results demonstrate the complementarity of different proteomic platforms". This is a diplomatic statement but has not really been demonstrated. One exercise to test this complementarity would be to predict BMI and other phenotypes allowing variable selection from both platforms together, maybe predicting the cis-pQTL variants using protein levels from both platforms in a linear model to check in which cases they carry complementary information, i.e. as $SNP \sim pSOMA + pOLINK + \text{covariates}$.

Line 164: Why were two sets of imputations used and then merged? Is it really OK to pick the variants with the highest info score? This could lead to incoherent representation of the LD structure.

Line 345: The authors write: "The associations of proteins with a range of other traits are shown in eFigures 10-19." These are relevant plots showing interesting results, but they are barely touched upon in the paper later on. There is in general a lack of supplementary tables. The figures are nice, but a reader might want to look up specific results. Please provide the individual protein ids, their match between the platforms, a comprehensive set of descriptors of their correlation, and data used to generate the figures, so that these can be used in future work to select proteins that are consistently measured between platforms, e.g. for studies that seek replication and have data from different platforms.

Line 336: "Of the 654 BMI-associated proteins in both OLINK and SomaScan-ANML, 510 (78.0%) showed directionally consistent (i.e. shared) associations with BMI". A fraction of 22% of proteins showing opposite directionality while being associated with BMI is quite high. This needs to be discussed in the paper.

Please provide a Venn diagram and the corresponding data in table form, summarizing which proteins the authors believe to be consistent between both platforms regarding (1) shared pQTLs, (2) direct correlation between platforms, and (3) association with non-genetic phenotypes, i.e. BMI. I would consider this a main result from the study, as the intersection of the Venn diagram would contain those proteins that are most likely orthologs between both platforms and that can be used interchangeably between cohorts that use different platforms.

Line 424: SOMAScan runs three batches, with different aptamers used in different dilution ranges. This needs to be addressed and explained. Were the dilution ranges used as explanatory variables?

Line 497: The authors claim "this is the first such study to be conducted in East Asians, and this has enabled the identification of pQTLs that were not previously reported in other ancestry populations", but where is the data supporting it?

Line 515: I disagree with the statement "mass spectrometry platforms, which remains the 'gold standard' for protein identification and quantification" – MS is certainly the gold standard for identification, but not for quantification, at least not in blood proteomics on a population scale. The abbreviations of IHD and NRI should be spelt out in the abstract.

Line 117: The authors indicate the number of proteins on the latest OLINK and SOMA platforms versions here. It would be important to also cite the numbers for the platform versions used in the paper in this context.

Line 318: "In contrast, for proteins with cis pQTLs identified in only one platform, only 3.1% showed evidence for colocalization with the other platform". Isn't that to be expected? Why mention it?

Ethics approval: Please specify who the "relevant" ethical research committees are.

Data sharing: Please address sharing of the GWAS summary statistics, ideally this would be done through the GWAS catalog.

Reviewer #3 (Remarks to the Author):

Reviewer Comments

The study by Wang et al provides the so far largest in sample comparison between the two currently most comprehensive plasma proteomic platforms in a non-European ancestry. They report only moderate correlations ($\rho \sim 0.2$) across $\sim 1.6k$ commonly targeted proteins with subsequent weak overlap in genetic analysis.

The study is well-performed (specific comments below) but mostly confirmative, a replicating what had been previously reported and I missed a significant conceptual advance beyond the sample size/look at some more proteins. We already know about technicalities being the main determinants for differences between platforms and the relevance of ethnicity for genetic analysis also remained unanswered. In general, in the absence of a gold-standard measurements of protein targets it is hard to make any more sensible contribution than what had already been done to address the issue of cross-platform inconsistencies. The authors might be better off combining this cross-platform paper with their discovery work based on the Olink assay. This in particular relevance, as the study remains descriptive without clear evidence for superiority of any platform.

Specific comments:

- Why is the overlap comparatively small between the assays, if $>5k$ are targeted?
- General bias by choosing an IHD cohort, that may favour associations with a selected subset of proteins possibly better measured by one but not the other technology.
- The authors report protein batched to be an important predictor of Spearman correlation, but those also differ by lab and not only targets proteins (Boston vs Uppsala).
- More details on fine-mapping are needed (SuSiE tends to perform poorly for very strongly associated loci) and did coloc consider multiple causal variants? Further, coloc is known for performing poorly for regions in which both traits have extremely strong associations, leading to very small credible sets that are unlikely to overlap even when protein measures align well. An additional LD-based overlap will certainly help. This may also improve the overall number of colocalising pQTLs, as correspondence between lead and secondary signals had already been reported.
- What is the effect of ambient temperature and time since last meal on protein values?
- What explains the higher number of trans-pQTLs for SomaScan, is this due to unspecific pleiotropic variants?
- Are there differences in protein characteristics for proteins w/ and w/o cis-pQTLs on either platform? Previous work showed evidence for overlap with eQTLs and benign PAVs.
- The comparison of phenotypic associations is somewhat misleading, since it is upfront unclear, what true-positive associations there might be, and a measure of recovery would be needed. Currently, the data presented adds little to what is already known. It is also somewhat circular, to report concordant associations with BMI for proteins correlating well between platforms. The same applies to conclusions about protein targets consistently associated with IHD. It is of interest whether there are any systematic factors explaining differences.
- The LASSO analysis is flawed (using significant proteins is already biasing towards overfitting and a clear split into test and independent validation set would be needed to demonstrate value of either proteomic assay, instead of overfitting by numbers of proteins) and NRI not a good measure of better performance (<https://link.springer.com/article/10.1007/s12561-014-9118-0>). It is also unclear what drives the improvement, if number of proteins or type of platform seems to not matter much.

Response to reviewers' comments

(Ref No.: NCOMMS-23-59940)

We thank the three reviewers for their helpful comments and suggestions. We have revised the manuscript, with new analyses, Figures and Tables, and an updated Supplement. We also updated our GWAS using an updated LD reference panel, and updated our risk prediction model using two different methods. The reviewers' verbatim comments are shown below in bold followed by our responses in non-bold text. Page and line numbers indicating where changes have been made refer to the new version of the manuscript with tracked changes.

REVIEWERS' COMMENTS

Reviewer #1

The authors of "Comparative studies of genetic and phenotypic associations for 2,168 plasma proteins measured by two affinity-based platforms in 4,000 Chinese adults" present a clear manuscript on important topics; proteophenomics and proteogenomics.

I have two main remarks:

A) The first one is about the chronology of the presentation. The authors announce 2168 protein matched by both assays but then go into 1694 proteins where both Olink and Soma are attempting a single reagent each. Then discuss the close to 500 where there is not a one to one match. I very strongly recommend that the authors present first an overall statistics of what is the main result for the 2168. Then they can go and split it. One way to count each protein once and to wait by the inverse of number of pairs. The data are currently presented in an artificial manner. Let's remember that even if protein names are unique, there is likely multiple proteoforms per protein (isoform, bound vs free, cleaved, un-cleaved).

Response: Following the reviewer's advice, we have now updated the manuscript by presenting the results for the overall 2,168 proteins (2,749 pairs) first, and then split them into one-to-one and non-one-to-one matched pairs. The median Spearman's rho was 0.29 for all 2,749 pairs, which did not change when using the weighted method suggested by the reviewer. This is close to the median rho = 0.26 when restricted to the 1,694 one-to-one matched pairs (page 7, lines 161-173).

We also agree that even for those one-to-one matched pairs, the platforms might still measure different proteoforms of the same protein. This point has now been highlighted in page 17, lines 418-423.

Related to that topic in the chapter, line 374, about the proteins with non one to one pairing they should take the opportunity to go say how often there is at least one pair over lets say 0.4 and when all are lower.

Response: As suggested, we performed an additional analysis and added this piece of information to the results section (page 11-12, lines 275-282). Of the 472 proteins targeted by multiple SOMAmers, 300 (63.6%) had at least one reagent pair with a Spearman's rho >0.4, while the rest all had reagent pairs with a Spearman's rho ≤0.4.

B) The second comment is more about results and points not currently in the manuscript that I think would greatly improve the manuscript. First, let me talk about my first impression and the one of readers when seeing a title on proteogenomics among Chinese: we all appreciate a greater ethnic diversity in genomic studies currently dominated by studies among European subjects. I missed to see an angle in the results about the Ethnic angel. Following that comment I have three suggestions and one extra point?

B1- I suggest that the authors attempt to compare LD class of pQTL in Chinese vs European. Is there any ancestry refinement going one or the other direction. OD the authors have nice examples?

B2- There is a number of GWAS of diseases in East Asian including Chinese. Can the authors make an attempt of colocalising pQTL(mainly *cis*) with disease association in the same ethnic group

B3- Do the authors can think of any diseases that is more common in Chinese and do the same as in B2

Currently the ethnic angle is limited to discussion. I suggest it to be main result section.

Response: We agree with the reviewer's view on the importance of the ethnic diversity in genomics research and appreciate their valuable insights.

In response to B1, as the paper's main focus is the comparison between OLINK and SomaScan, we did not expand on the comparison of pQTLs between East Asian and European populations, which has been covered by our previous analysis on the OLINK batch-1 proteins in this same sample [Ref 19]. In this study, we found that >30% of the identified credible sets of pQTLs did not overlap with those found in European populations. Our latest updated analysis incorporating both the first- and second-batch OLINK proteins showed the similar results. We have added this information to the discussion section and cited this article (page 14, lines 331-335).

To highlight the perspective on ethnic diversity, we have also moved the examples of *cis*-pQTLs for ALDH2 and PLA2G7 to our results section (page 9, lines 209-215), which are both very rare variants in European populations and thus not identified in previous pQTLs studies.

In response to B2 and B3, the utility of such East-Asian-specific pQTLs can be illustrated by our previous work, which employed the *cis*-pQTL for ALDH2 as a genetic instrument for alcohol intake in Mendelian Randomisation. We showed causal associations between genetically predicted alcohol intake and multiple diseases and mortality in Chinese men [Ref 20-22]. Similarly, we have also employed the *cis*-pQTL for PLA2G7 to conduct a phenome-wide association study, but did not find any associations with risks of vascular or non-vascular diseases in Chinese adults [Ref 25]. Those studies have now been discussed and cited in the discussion section (pages 14, lines 335-346).

**The following extra point is not a review point, more a mention of my surprise:
Checking the affiliation of authors, I was a surprised to see that only four of the around 25 individually listed authors where affiliated in China. I am aware of the CKB group last authorship.**

Response: For clarification, the authorship for this paper complied with the internal policy of China Kadoorie Biobank and the principle promoted by the Global Code of Conduct for Equitable Research Partnerships. Overall, more than half of the publications in CKB are led by authors affiliated with China. To recognise the contribution of the members of the CKB collaborative group, we will list all other members of the group in **Supplementary Note 1** of the published paper, which will be indexed in PubMed as collaborators.

**Overall, this is a very good paper, that I think would greatly benefit from
- presenting the overall comparison and not focus in abstract on the 1694 but go straight to the 2168 proteins
- investigating and presenting results on the strength of the paper which is the population.**

Response: We thank the reviewer for their positive feedback and valuable suggestions. We have updated the manuscript according to the advice, including describing the comparison based on all 2,168 proteins in the abstract, and highlighting the ethnic diversity perspective in the paper.

Reviewer #2

This paper presents a head-on comparison of the two major affinity proteomics platforms, SOMAscan and Olink, on a large number of samples of Chinese origin (N=4000). Comparison is performed by direct correlation and also by shared association with genetic and non-genetic traits. The paper addresses an important question as both platforms are increasingly used in large scale population studies, but previous smaller scale studies suggested that the agreement between both platforms is less than expected. Different reasons for these disagreements have been proposed, including differences in target binding of the affinity probes and detection limits. While this paper does not answer all of these questions, it advances our insights substantially. Most striking is the finding of a bimodal distribution of Spearman's rho correlation presumably identical protein targets, with a peak centered around 0 and another closer to 0.8. This should be emphasized and discussed further, rather than stating in the abstract "there was a modest between platform correlation (median rho=0.20)". The correlation between the platforms is actually quite good, but only on a smaller subset of protein targets.

Taken together this is a very relevant paper that will eventually be of great interest to the wider readership of Nature Communications. However, there are a number of points where the paper could be improved, both regarding clarity and comprehensiveness.

Specific points:

1. ANML versus non-ANML: To make the paper easier to read, use and discuss only one of the two normalization methods provided by SOMAscan (ANML or non-ANML) in the main text and move the comparison of ANML to non-ANML to the supplement (or remove it entirely). This is interesting but distracts from the main question. Removing one will also render the Figures more readable, as there will be less plots to show. Also decide from the start on log-scaling the SOMA data. There is no need to show that log-scaling works better, as the OLINK data is already on a log-scale.

Response: We thank the reviewer for their valuable suggestions and feedback. As suggested, we have now moved the results of SomaScan-ANML from the main text to the supplement and only kept the results based on SomaScan-non-ANML in the main text. This is because the OLINK data did not include the ANML step, which makes them more comparable with the SomaScan-non-ANML data, as reflected by their higher observational correlations.

As suggested, we have also removed the results comparing the SomaScan data before/after log transformation with OLINK from the manuscript.

2. Use Spearman correlations throughout for consistency (or Pearson, but not multiple methods).

Response: We understand the reviewer's concern about the use of both Spearman and Pearson correlations in the manuscript. Because Pearson's r assesses linear relationships and Spearman's rho assesses monotonic relationships, we chose to use both of them in the paper but in different contexts.

As the protein levels were measured by two different platforms and were originally provided on different scales, we used Spearman's rho to assess their correlation, since it is not affected by data transformation and this relationship could be non-linear. We used Pearson's r to assess the correlation between effect sizes, as the effect sizes were standardised across platforms and this relationship is largely linear (as shown in **Figure 3**).

For completeness, we also provided Pearson's r for the correlation between protein levels measured by OLINK and SomaScan in **Supplementary Figures 2 and 4**. We hope this will offer a more comprehensive assessment of the comparability between the two platforms.

3. Matching of proteins between the platforms: The authors use UniProt ids to identify orthologous targets and limit the major part of the study to proteins with a 1-1 match. However, I would also be interested in knowing which proteins actually are the best mutual hits on a pure correlation basis, assuming of course that 1-1 matches should also be their best mutual best correlation hit. Is that always the case? Are there exceptions? This would also allow to address the question as to which of the multiple SOMA targets are actually orthologs of the OLINK targets, which could then be added to the general analysis. By mutual best hit I understand pairs of proteins where one is the highest correlate of the other and vice versa (based on an all-against-all correlation between platforms). If a protein is only present on one platform, then its mutual best hit can of course be another protein that is related to the same pathway (which is also of interest). I believe a table of mutual best hit correlations would be of great interest. The concept is borrowed from the definition of orthologous genes between bacteria. I would encourage the authors to proceed to such an analysis, as it validates the protein match based on UniProtID. It may also identify binders that target the wrong protein, as happened on the SOMAscan platform in the past when some aptamers were developed on a wrongly labeled protein sample (also a possible reason for lack of correlation).

Response: As suggested, we have now calculated the correlations of all possible reagent pairs between OLINK and SomaScan for all 2,168 proteins included in the paper. The results of this analysis are shown in **Supplementary Note 2**.

As shown in the heatmap (**Supplementary Figure 6**), although the correlations seem randomly distributed overall, a faint diagonal line representing the UniProt-matched pairs is still visible. This is consistent with our finding of the modest correlation between the UniProt-matched pairs.

Of the 2,168 proteins, 739 pairs were mutual best hits between OLINK and SomaScan (where the reagents were each other's best hit). This could be a result of reagents binding different proteoforms of the same protein, or could indicate the binding of reagents on non-target proteins. However, of the 739 pairs of mutual best hits, 706 were also matches based on UniProt IDs, suggesting that our matching method based on UniProt IDs is a valid approach.

4. What is the rationale of using ambient temperature at sample collection as co-variate? How strong is that signal? The reason I am asking is that we saw a strong association with what Somalogic calls "cell abuse" in PCAs. One proxy for this kind of cell abuse is HSP90, and adding HSP90 as a covariate significantly improved many associations. The hypothesis is that some of the proteins originate from white blood cells, and then that white blood cells are more frequently spilling their content after collection at higher ambient temperatures. It would also be interesting to run a PCA on the proteins and investigate whether one of the major components correlate with HSP90, other proteins labelled as markers for "cell abuse" by Somalogic, or ambient temperature. More generally, it would be a good idea to correlate the first PCs from both platforms to check for shared general variance, be it technical or biological.

Response: In both platforms a number of proteins were shown to be associated with ambient temperature, as shown in **Supplementary Figures 20 to 25**. However, no significant associations were found between HSP90 and ambient temperature in either OLINK or SomaScan in our study. This is not surprising, as HSP90 might be more related to the response to stressful conditions (such as heat shock), instead of ambient temperature that is more stable and general. However, we did find significant associations between HSPB1 and ambient temperature in both platforms. Given the number of significant associations observed, the effect of ambient temperature on protein levels warrants further research, which will be covered by our future studies that involve more detailed analyses on this topic.

In the data we received from SomaScan, no proteins were marked as 'cell abuse'. This might be an update in the later versions of the SomaScan assay, where additional standardisation procedures have been incorporated to account for such influence from technical factors.

Following the reviewer's advice, we performed PCA on protein levels in both platforms and checked the associations between ambient temperature/time since last meal and the first PC of each platform (**Supplementary Note 3; Supplementary Figure 19**). We found that PC1 of SomaScan-ANML was significantly associated with ambient temperature, while PC1 of OLINK, SomaScan-ANML, and SomaScan-non-ANML were all significantly associated with time since last meal. This suggests the general influence of environmental/technical factors on proteomic measurements. The results of associations between individual proteins and ambient temperature/time since last meal are included in **Supplementary Data 3**.

5. Please provide individual scatter plots for all proteins as a supplementary PDF. It would be important to show the raw data at some point without readers having to write a biobank application first (consider sharing the proteomics data outside a biobank application, if that is possible – final SOMA and OLINK datasets are small enough to be provided as Supplementary Tables).

Response: Unfortunately, we are unable to include all scatter plots and raw data in the supplement, as this would require more than 2,000 individual plots and two 4000*7000 and 4000*3000 tables. However, we have now provided the summary results of all OLINK-SomaScan pairs as **Supplementary Data**. Results of correlation coefficients, colocalisation, and proteomic associations are all included and can be used by future researchers as reference.

6. This paper comprises a full pQTL study, but little data is actually shown. What are the plans for sharing the GWAS summary statistics? How do the pQTLs compare to published pQTL studies in Europeans? Further analysis of these might be the topic of another paper, but since the pQTLs are central to the comparison, this needs to be covered to a certain extent.

Response: We will include detailed results of the two pQTL studies in our two future GWAS papers on OLINK and SomaScan proteins, of which the summary statistics will be made available through GWAS catalog and CKB "PheWeb" browser. The preprint of our pQTL analysis on the first-batch OLINK proteins is already available on medrxiv, and we now cited this preprint in our updated manuscript [Ref 19]. The pQTL studies on the second-batch OLINK proteins and SomaScan proteins will be updated and published in due course.

7. The authors state that "The results demonstrate the complementarity of different proteomic platforms". This is a diplomatic statement but has not really been demonstrated. One exercise to test this complementarity would be to predict BMI and other phenotypes allowing variable selection from both platforms together, maybe predicting the *cis*-pQTL variants using protein levels from both platforms in a linear model to check in which cases they carry complementary information, i.e. as $SNP \sim pSOMA + pOLINK + \text{covariates}$.

Response: The word 'complementarity' intended to highlight that each platform has its own advantages and may offer additional information in comparison, directly or indirectly, with each other. Following the reviewer's advice, we have added an additional analysis using all 1,694 overlapping proteins from **both** platforms to predict incident IHD. Surprisingly, using proteins from **both** platforms did not seem to significantly improve the model performance compared with the model including proteins from only **one** platform. This suggests that using a set of core proteins from one platform alone may be sufficient to achieve optimal performance of risk prediction. We have now added this analysis to the Results section (page 11, lines 259-261) and also discussed the finding in the revised manuscript (page 15-16, lines 372-381). Based on this, we also changed our wording in the abstract to 'the results demonstrate the strengths of each proteomic platform and should inform assay selection in future studies'.

8. Line 164: Why were two sets of imputations used and then merged? Is it really OK to pick the variants with the highest info score? This could lead to incoherent representation of the LD structure.

Response: We employed two imputation servers in our genetic studies considering the unique genetic architecture of the Chinese population. This was done to combine the strengths of both

servers, as TOPMed has a much larger sample size and has better coverage of variants of higher frequency, while WBBC is considered better for rarer and Chinese-specific variants. We have observed for more common SNPs, TopMed imputation is of better quality, while WBBC is better at imputing rare variants.

We understand the reviewer's concern about LD, but so far we haven't observed any LD structure incoherence in our imputed genetic data. For the rare variants imputed by WBBC, we found that variants in high LD with those rare variants and presumably in the same haplotype were simultaneously included in our imputed data.

9. Line 345: The authors write: “The associations of proteins with a range of other traits are shown in Supplementary Figures 10-19.” These are relevant plots showing interesting results, but they are barely touched upon in the paper later on. There is in general a lack of supplementary tables. The figures are nice, but a reader might want to look up specific results. Please provide the individual protein ids, their match between the platforms, a comprehensive set of descriptors of their correlation, and data used to generate the figures, so that these can be used in future work to select proteins that are consistently measured between platforms, e.g. for studies that seek replication and have data from different platforms.

Response: We have now included all information suggested by the reviewer in **Supplementary Data**, including all reagent pairs and their correlation coefficients, colocalisation results, and results of their associations with the selected traits. We hope this information is now sufficient and will be beneficial to future research.

10. Line 336: “Of the 654 BMI-associated proteins in both OLINK and SomaScan-ANML, 510 (78.0%) showed directionally consistent (i.e. shared) associations with BMI”. A fraction of 22% of proteins showing opposite directionality while being associated with BMI is quite high. This needs to be discussed in the paper.

Response: Following the reviewer's advice, we further checked the BMI-associated proteins that showed opposite directions between the two platforms, with further discussion in the revised manuscript (page 15, lines 354-360). Those proteins generally had low observational correlations (median Spearman's rho = 0.04), which could indicate proteoforms differentially associated with BMI, or false positives that arose due to technical differences between the platforms.

11. Please provide a Venn diagram and the corresponding data in table form, summarizing which proteins the authors believe to be consistent between both platforms regarding (1) shared pQTLs, (2) direct correlation between platforms, and (3) association with non-genetic phenotypes, i.e. BMI. I would consider this a main result from the study, as the intersection of the Venn diagram would contain those proteins that are most likely orthologs between both platforms and that can be used interchangeably between cohorts that use different platforms.

Response: As suggested, the Venn diagram has now been included in **Supplementary Figure 38** and discussed (page 17, lines 404-412). We found that proteins with high observational correlations and colocalising *cis*-pQTLs were more likely to yield shared associations for traits and disease risks, and findings on those proteins in future studies may be more generalizable across cohorts using different platforms.

12. Line 424: SOMAscan runs three batches, with different aptamers used in different dilution ranges. This needs to be addressed and explained. Were the dilution ranges used as explanatory variables?

Response: SomaScan used three serial dilutions (0.05%, 0.5%, and 20%) to achieve optimal detection of protein targets, and one SOMAmer is only present in one of the three dilutions, which is consistent across all participants. Since dilution is the same for all participants given a particular SOMAmer, we did not include it as an explanatory variable in our analysis. This information has been added to the methods section.

13. Line 497: The authors claim “this is the first such study to be conducted in East Asians, and this has enabled the identification of pQTLs that were not previously reported in other ancestry populations”, but where is the data supporting it?

Response: We have now added the examples of two East-Asian-specific pQTLs for ALDH2 and PLA2G7 to our results section to illustrate this point (pages 9, lines 209-215). Both pQTLs had low MAF in European populations, and thus were not identified in previous studies.

Moreover, as mentioned earlier, our previous GWAS analysis on the first-batch OLINK proteins revealed that >30% of the identified credible sets of pQTLs did not overlap with pQTLs found in European populations [Ref 19]. Our recent updated analysis incorporating both the first- and second-batch OLINK proteins showed the similar results. In our previous work, we have also employed the East-Asian-specific pQTLs for ALDH2 and PLA2G7 as genetic instruments in Mendelian Randomisation studies [Ref 20-22, 25]. To highlight the importance of ancestry diversity in genomics and proteomics research, those studies have now been discussed in the the revised manuscript (pages 14, lines 331-346).

14. Line 515: I disagree with the statement “mass spectrometry platforms, which remains the ‘gold standard’ for protein identification and quantification” – MS is certainly the gold standard for identification, but not for quantification, at least not in blood proteomics on a population scale.

Response: We thank the reviewer for their insights. We have now removed ‘quantification’ from this statement.

15. The abbreviations of IHD and NRI should be spelt out in the abstract.

Response: This has been updated in the abstract.

16. Line 117: The authors indicate the number of proteins on the latest OLINK and SOMA platforms versions here. It would be important to also cite the numbers for the platform versions used in the paper in this context.

Response: This has now been made clear in the introduction, results, and methods sections of the paper, and also included in **Figure 1**. In the current study, there were 2,923 reagents (mapped to 2,923 UniProt IDs) included in the OLINK Explore 3072 platform, and 7,301 SOMAmers targeting human proteins (mapped to 6,397 UniProt IDs) included in SomaScan Assay v4.1.

17. Line 318: “In contrast, for proteins with *cis* pQTLs identified in only one platform, only 3.1% showed evidence for colocalization with the other platform”. Isn’t that to be expected? Why mention it?

Response: Since colocalisation could capture shared signals even if no *cis*-pQTLs were identified in one of the platforms, we mentioned this for the completeness of our analysis. However, it is indeed expected that the proportion is low. We have now removed this sentence from our manuscript to avoid redundancy.

18. Ethics approval: Please specify who the “relevant” ethical research committees are.

Response: Ethical approval was obtained from the Ethical Review Committee of the Chinese Centre for Disease Control and Prevention (Beijing, China, 005/2004) and the Oxford Tropical Research Ethics Committee, University of Oxford (UK, 025-04). This has been updated in the revised manuscript (page 26, lines 618-621).

19. Data sharing: Please address sharing of the GWAS summary statistics, ideally this would be done through the GWAS catalog.

Response: We will include detailed results of the two pQTL studies in our two future GWAS papers on OLINK and SomaScan proteins, of which the summary statistics will be made available through GWAS catalog and CKB “PheWeb” browser. The preprint of our pQTL analysis on the first-batch

OLINK proteins is already available on medRxiv, and we now cited this preprint in our updated manuscript (page 14, lines 331-335) [Ref 19].

Reviewer #3

The study by Wang et al provides the so far largest in sample comparison between the two currently most comprehensive plasma proteomic platforms in a non-European ancestry. They report only moderate correlations ($\rho \sim 0.2$) across ~1.6k commonly targeted proteins with subsequent weak overlap in genetic analysis.

The study is well-performed (specific comments below) but mostly confirmative, a replicating what had been previously reported and I missed a significant conceptual advance beyond the sample size/look at some more proteins. We already know about technicalities being the main determinants for differences between platforms and the relevance of ethnicity for genetic analysis also remained unanswered. In general, in the absence of a gold-standard measurements of protein targets it is hard to make any more sensible contribution than what had already been done to address the issue of cross-platform inconsistencies. The authors might be better off combining this cross-platform paper with their discovery work based on the Olink assay. This in particular relevance, as the study remains descriptive without clear evidence for superiority of any platform.

Response: As recognised by Reviewers 1 and 2, we believe our study adds valuable insights into the current field of proteomics research, not only because of its significantly bigger sample size/more overlapping proteins, but also because it is the first study to directly compare the two platforms in the East Asian population. Moreover, we are also the first to provide a comprehensive assessment of their cross-platform comparability not only in pQTLs but also in observational correlations, phenotypic associations, and disease risk prediction, while previous studies only focused on one or two aspects of them.

We share the reviewer's concern about the lack of gold-standard measurements that could act as a reference to assess each platform's performance. However, using OLINK itself as a reference might not be ideal, since neither platform showed superior performance to the other across all the aspects, as suggested by the results of our current study. In theory, mass spectrometry would be the optimal choice as a gold-standard measurement, but its large-scale application in population-based studies remains challenging, and we have discussed this in the discussion section of the paper. Moreover, even without the presence of such a gold standard, we still identified a set of proteins that tend to produce replicable results across platforms, which will be useful for future studies using different platforms.

Specific comments:

1. Why is the overlap comparatively small between the assays, if >5k are targeted?

Response: In our current study, we employed the OLINK Explore 3072 and the SomaScan Assay v4.1 platforms. The SomaScan Assay v4.1 included 7,301 protein measurements in total, but the majority of them, usually in low abundance, were not included in OLINK Explore 3072 platform. However, of the 2,923 proteins included in the OLINK Explore platform, over 2/3 were actually also covered by the SomaScan Assay v4.1, which represents a substantial overlap. This has been made clear in the revised manuscript (page 6, lines 144-151).

2. General bias by choosing an IHD cohort, that may favour associations with a selected subset of proteins possibly better measured by one but not the other technology.

Response: We understand the reviewer's concern about the design of our case-subcohort study. To minimise potential biases, we conducted additional analyses using randomly selected subcohort participants only, and found that the observational correlations between the two platforms barely changed (**Supplementary Figure 5**). Due to lower statistical power, the number

of significant proteomic associations with traits decreased among subcohort participants, but the overall patterns across the two platforms remained similar (**Supplementary Figures 20 to 25**). Therefore, we believe that the results of our analyses in the overall sample were valid and comparable to those in the subcohort participants.

- The authors report protein batched to be an important predictor of Spearman correlation, but those also differ by lab and not only targets proteins (Boston vs Uppsala).

Response: We thank the reviewer for their advice. It is indeed possible and we have now added this point to the results section (page 7-8, lines 181-184).

3. More details on fine-mapping are needed (SuSiE tends to perform poorly for very strongly associated loci) and did coloc consider multiple causal variants? Further, coloc is known for performing poorly for regions in which both traits have extremely strong associations, leading to very small credible sets that are unlikely to overlap even when protein measures align well. An additional LD-based overlap will certainly help. This may also improve the overall number of colocalising pQTLs, as correspondence between lead and secondary signals had already been reported.

Response: We thank the reviewer for pointing out the limitations of using the SuSiE approach for colocalisation. Indeed, SuSiE's performance might be unreliable for very strong signals. Therefore, following the reviewer's advice, we have changed our analytical approach and re-performed the colocalisation analysis using coloc.abf (colocalisation under the single causal variant assumption) for *cis*-pQTLs discovered in at least one platform. Using the new approach, we found more proteins that had evidence for colocalisation of *cis*-pQTLs between the two platforms, which has been updated in the revised manuscript (page 8, lines 194-200, and page 23, lines 546-549).

Following the reviewer's advice, we also calculated the r^2 for each protein between the sentinel variants discovered in OLINK and SomaScan if their *cis*-pQTL regions overlapped, and the result was consistent with the colocalisation analysis, which has been updated in the revised manuscript (page 8, lines 100-205). Of the 417 proteins (428 sentinel pairs) with *cis*-pQTLs identified in both platforms, the mean r^2 was 0.75 for all sentinel variant pairs for each protein. There were 307 proteins that had at least one pair of sentinel variants with an $r^2 > 0.6$.

4. What is the effect of ambient temperature and time since last meal on protein values?

Response: Please see our response to Reviewer 2's 4th specific comment.

5. What explains the higher number of trans-pQTLs for SomaScan, is this due to unspecific pleiotropic variants?

Response: Indeed, SomaScan identified slightly more proteins with trans-pQTLs than OLINK but the difference between the two platforms is modest. There are several possible explanations, including non-specific binding of reagents or unspecific pleiotropic variants, which have now been discussed in the revised manuscript (page 14, lines 327-330).

6. Are there differences in protein characteristics for proteins w/ and w/o *cis*-pQTLs on either platform? Previous work showed evidence for overlap with eQTLs and benign PAVs.

Response: As the current study focuses on the comparison between platforms, we did not compare the characteristics of proteins with and without *cis*-pQTLs within each platform. However, in Boruta feature selection, we included a list of protein characteristics as features to explore if any of them were predictive of observational correlations (**Supplementary Tables 1 & 2**). However, none of those features were confirmed by Boruta, suggesting that technical factors had more influence on the concordance between platforms than protein characteristics. In our future papers that focus specifically on the pQTLs of each platform, we will explore further the effect of eQTLs and PAVs on the identification of pQTLs.

7. The comparison of phenotypic associations is somewhat misleading, since it is upfront unclear, what true-positive associations there might be, and a measure of recovery would

be needed. Currently, the data presented adds little to what is already known. It is also somewhat circular, to report concordant associations with BMI for proteins correlating well between platforms. The same applies to conclusions about protein targets consistently associated with IHD. It is of interest whether there are any systematic factors explaining differences.

Response: We acknowledge that it remains unclear as to which of the associations between proteins and traits were true positives. However, we would like to highlight that for those proteins with high observational correlations, they were also more likely to yield consistent phenotypic associations across platforms, as shown in **Figures 3e and 3f**. Such findings will be helpful for future research to assess the generalisability of their findings on a particular protein, especially when different proteomic platforms are used by different studies. We have also added a Venn (**Supplementary Figure 38**) to illustrate this idea, as described in the discussion section (page 17, lines 404-412).

As indicated by our results of Boruta feature selection, assay-related technical factors play an important role in explaining the differences in observational correlations between the two platforms, which can further influence the association results. As an example, we further investigated the proteins that were significantly associated with BMI in both platforms but with opposite directions of their effects. Our analysis revealed that those proteins generally had low observational correlations (median Spearman's $\rho = 0.04$), which could indicate false positives that arose due to technical differences between the platforms. Alternatively, they might also suggest different proteoforms captured by different assay platform that were differentially associated with BMI. We have added this to the discussion section of the revised manuscript (page 15, lines 354-360).

8. The LASSO analysis is flawed (using significant proteins is already biasing towards overfitting and a clear split into test and independent validation set would be needed to demonstrate value of either proteomic assay, instead of overfitting by numbers of proteins) and NRI not a good measure of better performance (<https://link.springer.com/article/10.1007/s12561-014-9118-0>). It is also unclear what drives the improvement, if number of proteins or type of platform seems to not matter much.

Response: We understand the reviewer's concern about our risk prediction analysis. We have now updated our risk prediction models and provided a more detailed description of this analysis in the methods section of the revised manuscript. For clarification, all analyses were performed using 10-fold cross-validation to avoid overfitting.

We recognised that using the FDR-significant proteins might introduce issues of overfitting. Hence, we also included the results using all 1,694 one-to-one matched overlapping proteins, which would not be influenced by such problems. Moreover, as the primary aim of the current study is to compare the two platforms, any overfitting issues would affect both platforms similarly. Therefore, we believe our comparison between the two platforms is fair and valid.

We thank the reviewer for pointing out the potential issue of using NRI as a measure of prediction improvement. To provide a more comprehensive overview of the performance of risk prediction models, we also included the C-statistic of each model, and calculated NRI using both decile-based (**Figure 4 and Supplementary Figure 30**) and category-free (**Supplementary Figure 31**) methods (pages 24-25, lines 568-587). We hope that there is now sufficient information for our readers to assess the performance of different models based on those indices. Nevertheless, adding proteins to conventional risk factors increased C-statistic and yielded positive NRIs, supporting their utility in risk prediction for IHD.

Risk prediction models using the FDR-significant proteins yielded similar NRI and C-statistics compared to models using all 1,694 proteins. This suggests that the core proteins significantly associated with IHD play an important role in risk prediction, while adding additional proteins barely improved the model performance. We have also undertaken an additional analysis using protein measurements from both platforms, but this also did not significantly improve the model performance (page 11, lines 259-261). Taken together, this suggests that using a set of core proteins from one platform alone may be sufficient to achieve optimal performance of risk

prediction, at least for IHD. This point has now been made clear in the discussion section of the revised manuscript (page 15-16, lines 372-378).

REVIEWER COMMENTS

Reviewer #1 (Remarks to the Author):

The authors answered all my questions and adapted the manuscript accordingly.
I have no further comments

Reviewer #2 (Remarks to the Author):

The authors responded to most of the points I raised, except for the following:

Please provide the correlation data for the 739 mutual best hit pairs in a separate table, as a resource that shows orthology between the platforms on the correlation level.

I understand that this is not a full GWAS paper and that full pQTL summary stats are provided elsewhere. However, as pQTLs are also central to this analysis, summary statistics for the key pQTLs should be provided in table format with this paper (i.e. the ones used for Supplementary Figure 11).

Reviewer #2 (Remarks on code availability):

I don't think that a review at the code level is useful here. Without data it cannot be tested.

Reviewer #3 (Remarks to the Author):

Reviewer comments

I appreciate the efforts by the authors to address my concerns, and while some have been well addressed most remain somewhat unanswered. My general issue still is that more needs to be done to derive conclusion that haven't been already demonstrated by others. The authors have a lot of data to investigate this in much more detail.

Rebuttal

- 1) I still don't understand the rationale, why a comparison of the two platforms among East Asian participants provides any more value than in any other population. Differences will be driven mostly by technical factors, as the authors also demonstrate. Even if there are genuine different biological factors affecting plasma protein levels, the technology of how we measure those samples will not shed light much light or identify those when not compared to findings in other publications. It is further simply untrue, that authors claim to be the first to compare both technologies with respect to observational correlations (see for example <https://www.nature.com/articles/s41586-023-06563-x>; even if this was not done head-to-head).
- 2) I am not quite happy with the response to R3 comment 6. My concern has been, that poor correlation might be explained by platform-specific PAVs. From what I understand, this has not been included in the feature selection procedure.
- 3) R3 comment 7: The argument is still circular, why would a protein that is well measured by both assays give raise to different phenotypic associations. The authors are, as they claim, uniquely suited to understand differences across both platforms. Why is not possible to systematically investigate why protein associations with phenotypes differ. What makes those proteins special, what platforms recovers more true biological signals? This is needed to provide at least some advancement to the field that goes beyond being 'largest'. For example, how do the authors explain that more Olink proteins have cis-pQTLs and are associated with IHD, but fewer are associated with BMI? I think the

authors still fall short of demonstrating how their results inform assay selection in the future. We can simply not afford as a scientific community to spend massive amounts of (public) money, just because we still don't understand what technology to use.

4) R3 comment 8: 10-CV does not per se mitigate overfitting, it needs a clear split into training (that includes another splitting for repeated feature selection) and validation. At best, and since the authors claim to have the largest study, an another set for model optimization would be needed. I also disagree with the comment that overfitting will not matter when comparing both platforms. On the contrary, having artificially well-performing models will hide any underlying differences in both platforms. The NRIs reported in the abstract should also be replaced by more truthful measures, also following recent work (PMID: 38546334).

Manuscript

1) The authors need to tone down much of the claims they made throughout. For example, they start with saying that they compare 2,168 proteins, just to learn later on that only ~1.6k are eventually compared.

2) I appreciate that this has been done based on a comment from a reviewer, but recent work by deCODE people actually provided evidence that for genetic analysis the ANML normalization of SomaScan data is better, and choosing the normalization just to improve correlation with another assay is a poor choice. Both technologies are fundamentally different in the final readout of protein 'intensities' and hence require different normalisation steps. As genetic discovery is a key result here, the authors may want to reconsider their approach.

3) Line 209 onwards is one of the few references to ancestry-specific findings, is it only surprising to me, that both seem to map to PAVs, and hence might be problematic for affinity assays?

4) Line 252: How informative is a protein model that performs worse than a conventional risk factor model (c-index: 0.85 vs 0.82)? How have those c-indices been derived anyways? The authors should also stay away from suggestions such as in line 261 onwards given the obvious weaknesses of the study design and NRI itself.

5) Line 331 onwards, the reported 30% non-overlapping credible sets are likely a result of poor methodology rather than a true ancestral-specific findings. It adds to the attitude of the authors to overinterpret their findings. The example they give also refers to a PAV, which is particularly bad for MR with affinity-based proteomics as we cannot distinguish whether it truly affects plasma (or tissue) levels or simply how well the assay binds

6) Line 358: I may repeat myself, but these arguments are circular. If both assay do what they should do, measuring the protein target reliably, they will correlate and will give the same biological association. This is not a scientific achievement, but logic. The authors need to dive into the differences, if they really aim to provide guidance.

7) Line 377: Please stay away from statements like 'optimal performance'. While both assays measure a large number of proteins, they are highly selective and for most(!) there is no valid proof that they really circulate at stable levels in plasma.

Response to reviewers' comments

(Ref No.: NCOMMS-23-59940)

We thank the two reviewers (2 & 3) for their additional comments and suggestions. We have further revised the manuscript, with new analyses, Figures and Tables, and an updated Supplementary Information and Data to address or clarify the issues raised. In particular, we have included additional analyses on protein-altering variants and further explored the factors that might explain the discordant findings between platforms. We have also revised the section on risk prediction to offer a clearer description of the method and a more balanced assessment between platforms. The reviewers' verbatim comments are shown below in bold followed by our responses in non-bold text. Page and line numbers indicating where changes have been made refer to the new version of the manuscript. In the revised manuscript, the new changes are highlighted in **turquoise** and the old changes (in the previous revision) are highlighted in **yellow**.

REVIEWER COMMENTS

Reviewer #2

The authors responded to most of the points I raised, except for the following:

1) Please provide the correlation data for the 739 mutual best hit pairs in a separate table, as a resource that shows orthology between the platforms on the correlation level.

Response: Done. We have now provided the list of mutual best hits (based on both ANML and non-ANML SomaScan data) in **Supplementary Data 2**.

2) I understand that this is not a full GWAS paper and that full pQTL summary stats are provided elsewhere. However, as pQTLs are also central to this analysis, summary statistics for the key pQTLs should be provided in table format with this paper (i.e. the ones used for Supplementary Figure 11).

Response: Done. We have now included the list of pQTLs used as examples (ALDH2, PLA2G7, GHR, WFDC2, and TNC) in the study in **Supplementary Data 4**.

3) I don't think that a review at the code level is useful here. Without data it cannot be tested.

Response: CKB is committed to making the data available to the wider scientific community after a period of exclusive use by study investigators (see CKB data sharing policy at <https://www.ckbiobank.org/data-access>). The proteomics summary data (e.g. protein pQTLs) will be made available after manuscript publication through the relevant journal site and/or a newly developed CKB PheWeb (to be operational soon). Code used for comparative analyses in the current study can be accessed through: <https://github.com/baihanwang211/proteomics-public>.

Reviewer #3

I appreciate the efforts by the authors to address my concerns, and while some have been well addressed most remain somewhat unanswered. My general issue still is that more needs to be done to derive conclusion that haven't been already demonstrated by others. The authors have a lot of data to investigate this in much more detail.

Response: We thank the reviewer for further helpful comments to improve the quality of our manuscript. As suggested, we have now added more in-depth analysis to the revised manuscript, including analyses on protein-altering variants (PAVs) and additional factors that might explain the discordant findings between Olink and SomaScan. We have also provided a more detailed description of the risk prediction models, and

edited the manuscript to offer an unbiased comparison between the risk prediction models derived from the two platforms.

1) I still don't understand the rationale, why a comparison of the two platforms among East Asian participants provides any more value than in any other population. Differences will be driven mostly by technical factors, as the authors also demonstrate. Even if there are genuine different biological factors affecting plasma protein levels, the technology of how we measure those samples will not shed light much light or identify those when not compared to findings in other publications. It is further simply untrue, that authors claim to be the first to compare both technologies with respect to observational correlations (see for example <https://www.nature.com/articles/s41586-023-06563-x>; even if this was not done head-to-head).

Response: Our study is the first study that directly compares Olink and SomaScan in an East Asian population. As recognised by two other reviewers, we believe that the involvement of non-European populations has its own value, as the two platforms were primarily developed and compared in European populations and there is good evidence that genetic ancestry can impact protein concentrations [Ref 18]. Although previous studies showed moderate observational correlations between the platforms in European populations, it remained unclear if such findings could be generalizable to other populations, where there are now increasing enthusiasms for using different proteomics assays in clinical and population research. Furthermore, due to differences in allele frequencies, pQTLs identified in different populations may also differ (as demonstrated by our analysis comparing Olink pQTLs in UKB and CKB [Ref 20]), and this could further influence the concordance of pQTLs between Olink and SomaScan in different populations. We have now modified the introduction section to further clarify the study rationale (page 5, lines 124-127), and hope this has now been made clearer to the readers.

Although similar studies with direct comparisons have been published [Ref 13-17], they either compared the two assays in two different samples (which made the comparison indirect and biased by differences between datasets) or included fewer participants/proteins (which limited their power and breadth of coverage). We thank the reviewer for drawing our attention to the recent paper by Eldjarn et al, which was cited [Ref 17] in the original manuscript. In this paper, the main comparison was made based on two different datasets from UKB and deCODE, while the Spearman correlation was estimated in a smaller sample of 1,514 participants (i.e. <40% of the sample size in our study). Hence, apart from being in a non-European population, another strength of our study is that it is the largest study so far that directly compared the two proteomic platforms in identical blood samples. To avoid confusion, we have revised the statements in page 5, lines 117-130 to make it clear that we were comparing with other studies conducted in the same sample.

2) I am not quite happy with the response to R3 comment 6. My concern has been, that poor correlation might be explained by platform-specific PAVs. From what I understand, this has not been included in the feature selection procedure.

Response: Following the reviewer's advice, we have now annotated the *cis*-pQTLs identified in both platforms and checked if they were PAVs or in LD ($r^2 > 0.8$) with PAVs. We found that proteins with PAV *cis*-pQTLs had similar observational correlations to those with non-PAV *cis*-pQTLs (page 9, lines 213-230). We then used logistic regression to test if PAVs (as well as observational correlations and key technical factors) had effects on *cis*-pQTLs being Olink-specific, SomaScan-specific, or non-colocalising between platforms, using colocalising *cis*-pQTLs as the reference group. We found that SomaScan PAV *cis*-pQTLs were more likely to not colocalise with the Olink *cis*-pQTLs identified for the same protein, suggesting potential epitope effects. Those results are shown and discussed in page 9, lines 213-230; pages 18-19, lines 443-463.

3) R3 comment 7: The argument is still circular, why would a protein that is well measured by both assays give raise to different phenotypic associations. The authors are, as they claim, uniquely suited to understand differences across both platforms. Why is not possible to systematically investigate why protein associations with phenotypes differ. What makes those proteins special, what platforms recovers more true biological signals? This is needed to provide at least some advancement to the field that goes beyond being 'largest'. For example, how do the authors explain that more Olink proteins have *cis*-pQTLs

and are associated with IHD, but fewer are associated with BMI? I think the authors still fall short of demonstrating how their results inform assay selection in the future. We can simply not afford as a scientific community to spend massive amounts of (public) money, just because we still don't understand what technology to use.

Response: In our original manuscript, we did identify various factors that could likely contribute to observational correlations between Olink and SomaScan using Boruta feature selection. Following the reviewer's advice, we have now conducted additional analyses to further explore these and other factors more comprehensively. We firstly categorised proteins into four groups based on whether they had Olink-specific, SomaScan-specific, distinct, or shared findings on *cis*-pQTLs, BMI associations, and IHD associations. We then used logistic regression to test if observational correlations, key technical factors, or PAVs could explain any of the observed discordant findings, using proteins with shared findings as the reference group. Overall, we found that proteins with lower observational correlations, with lower abundance, or with more samples below LOD or more samples with QC warnings in Olink were more likely to produce platform-specific or distinct findings between platforms (**Supplementary Figures 13, 20, and 32**). *Cis*-pQTLs that were PAVs or in LD with PAVs might explain non-colocalising *cis*-pQTLs between the platforms, although they were also more likely to yield BMI associations that were shared by both platforms. Those additional analyses have now been added and discussed in page 9, lines 213-230; pages 10-11, lines 254-260; page 12, lines 280-284; pages 18-19, lines 431-463.

4) R3 comment 8: 10-CV does not per se mitigate overfitting, it needs a clear split into training (that includes another splitting for repeated feature selection) and validation. At best, and since the authors claim to have the largest study, another set for model optimization would be needed. I also disagree with the comment that overfitting will not matter when comparing both platforms. On the contrary, having artificially well-performing models will hide any underlying differences in both platforms. The NRIs reported in the abstract should also be replaced by more truthful measures, also following recent work (PMID: 38546334).

Response: For clarification, the development of risk prediction model involved 10-fold cross-validation, for which the overall data were split into 10 folds, and each time 9 folds were used as the training dataset while 1 fold was used as the test dataset. For hyperparameter optimisation, the lamda (regularisation) hyperparameter was selected using another 10-fold cross-validation within the training dataset each time. In this way, the training and test datasets were completely independent of each other, and the hyperparameter optimisation was also only conducted within the training dataset separately from the test dataset. Moreover, we also repeated this procedure 5 times to minimise noise introduced by different data splits. We believe this approach with clear splitting and independence of training and test datasets should sufficiently avoid or minimise overfitting. We have now updated the methods section to make this clearer to the readers (pages 26-27, lines 629-652).

However, we acknowledge that the IHD risk-prediction models built on the FDR-significant proteins might have introduced bias in protein selection, compared to the models using all 1,694 one-to-one matched overlapping proteins. Therefore, we have changed our abstract and manuscript to focus on the models using all overlapping proteins (page 3, lines 79-81; page 12, lines 285-300), which should offer an unbiased assessment of model performance between Olink and SomaScan. The models with FDR-significant proteins have been moved to **Supplementary Information**.

Manuscript

1) The authors need to tone down much of the claims they made throughout. For example, they start with saying that they compare 2,168 proteins, just to learn later on that only ~1.6k are eventually compared.

Response: Our analyses did cover all 2,168 proteins. However, due to the complexity of non-one-to-one reagent binding we chose to focus our main results on one-to-one matched pairs, with analyses on non-one-to-one matched pairs presented separately in the manuscript (page 13, lines 308-315). We think this approach is appropriate and would avoid potential confusion to our readers. However, we recognise that

the current title might not fully reflect the different emphasis on one-to-one matched and on non-one-to-one matched pairs, and would be happy to change the title to 1,694 proteins if it is deemed appropriate by the editorial team.

2) I appreciate that this has been done based on a comment from a reviewer, but recent work by deCODE people actually provided evidence that for genetic analysis the ANML normalization of SomaScan data is better, and choosing the normalization just to improve correlation with another assay is a poor choice. Both technologies are fundamentally different in the final readout of protein 'intensities' and hence require different normalisation steps. As genetic discovery is a key result here, the authors may want to reconsider their approach.

Response: We appreciated the reviewer's concern over the choice between ANML and non-ANML for comparison. It is indeed true that SomaScan-ANML identified more pQTLs than SomaScan-non-ANML in GWAS in both our study and previous studies by deCODE. We have now further highlighted this in page 17, lines 400-404 to give the readers sufficient information on the comparison between ANML and non-ANML.

We chose to use SomaScan-non-ANML data for main analyses, because the aim of the paper is to assess the comparability between the two platforms covering not only genetic discovery (as was done primarily in previous papers), but also phenotypic associations and risk prediction. While Olink and SomaScan employ different methods throughout their data processing pipelines, we believe that data normalisation based on external references (as done for SomaScan-ANML but not for Olink) is a fundamental one, as it can have a significant impact on data distribution. Moreover, it is unclear if the external references used by SomaScan would be suitable for the East Asian population and what impacts they might have on the resulting data. Nevertheless, we have presented all the results based on SomaScan-ANML data in **Supplementary Information**, which will enable readers to assess their concordance with Olink separately.

3) Line 209 onwards is one of the few references to ancestry-specific findings, is it only surprising to me, that both seem to map to PAVs, and hence might be problematic for affinity assays?

Response: We thank the reviewer for pointing out that both *cis*-pQTLs used as examples here are PAVs. It is indeed possible that the *cis*-pQTLs identified here only alter protein structure instead of protein concentrations, thus leading to false positives and epitope effects. However, there is evidence that the sentinel *cis*-pQTL variant (rs671) identified in our study for ALDH2 affects ALDH2 concentrations (by reducing half-life, measured through immunoblotting) in human liver tissues [Ref 28 and 29], and our finding on its association with alcohol intake as well as multiple diseases and mortality in CKB further demonstrated its functional implications. Similarly, the sentinel variant (rs76863441) for PLA2G7 has also been reported to be associated with the concentration of Lp-PLA2 (measured by a different immunoassay, diaDexus PLAC Test ELISA Kit) [Ref 30]. We have now updated the manuscript to discuss this in more detail (page 16, lines 381-391).

4) Line 252: How informative is a protein model that performs worse than a conventional risk factor model (c-index: 0.85 vs 0.82)? How have those c-indices been derived anyways? The authors should also stay away from suggestions such as in line 261 onwards given the obvious weaknesses of the study design and NRI itself.

Response: Indeed, the models with proteins alone did not outperform the model with conventional risk factors (which included strong risk factors such as age) in the current study. However, the aim of the study was to assess the comparability of the two platforms, instead of the predictive ability of proteomics in general. Therefore, we only built the risk prediction models using 1,694 overlapping proteins with one-to-one matched pairs between the two platforms, and were able to compare the performance of these models using each platform with proteins alone or in addition to conventional risk factors. Furthermore, in a separate report (currently under review), we have demonstrated that a protein-based risk prediction model, developed using all measured proteins (not just those overlapping), outperformed the model with conventional risk factors, supporting its clinical utility.

We have provided a more detailed description of the risk prediction models in pages 26-27, lines 629-652. The C-statistics were calculated as per the case-cohort design, of which more information can be found in

Ref 62. In brief, informative pairs were divided into case-case pairs and case-control pairs, and case-control pairs were weighted as those pairings were underrepresented in the case-cohort design.

As mentioned above, we have also changed the way we present and interpret the results of our risk prediction models, with the main focus on the models using 1,694 overlapping proteins (instead of FDR-significant proteins) to avoid bias. Moreover, we have now used both C-statistics and NRI to compare model performance (page 3, lines 79-81; page 12, lines 285-300; page 14, lines 333-334; pages 17-18, lines 417-425).

5) Line 331 onwards, the reported 30% non-overlapping credible sets are likely a result of poor methodology rather than a true ancestral-specific findings. It adds to the attitude of the authors to overinterpret their findings. The example they give also refers to a PAV, which is particularly bad for MR with affinity-based proteomics as we cannot distinguish whether it truly affects plasma (or tissue) levels or simply how well the assay binds

Response: For clarification, the same Olink Explore assay was used to measure plasma proteins in both CKB and UKB, which showed high reproducibility between replicated samples in previous research including UKB [Refs 12 & 13], despite its moderate correlation with the SomaScan platform. Therefore, we believe the differences between pQTLs found in CKB and UKB can be mainly attributed to ancestral differences instead of technical factors. However, we acknowledge that technical differences between studies could also influence pQTL results, and we have added this to our discussion section (page 15, lines 366-370).

For discussion on PAVs and the examples used in the current paper, please refer to our response to comment 3) above. However, to highlight the potential pitfalls of using PAVs in MR studies, we have added a word of caution to page 16, lines 389-391.

6) Line 358: I may repeat myself, but these arguments are circular. If both assay do what they should do, measuring the protein target reliably, they will correlate and will give the same biological association. This is not a scientific achievement, but logic. The authors need to dive into the differences, if they really aim to provide guidance.

Response: Please refer to our previous response to R3 comment 7 on the rebuttal letter.

7) Line 377: Please stay away from statements like 'optimal performance'. While both assays measure a large number of proteins, they are highly selective and for most(!) there is no valid proof that they really circulate at stable levels in plasma.

Response: For clarification, we used the word 'optimal performance' to refer to the performance of the risk prediction models, rather than the performance of assays in measuring protein levels. However, given that this might cause some confusion to the readers, we have now removed this phrase from the manuscript (page 17, lines 419-422).

REVIEWERS' COMMENTS

Reviewer #3 (Remarks to the Author):

Reviewer comments

Wang et al. did a considerable revision of their work, that improved the overall accessibility of the paper, including some additional interesting observations. However, I still have concerns of what the study adds over and above of what has been published. I clearly see the relevance of providing a comprehensive comparison between both platforms, but this has been done at considerable scale as outlined in my previous comments. Most importantly, the authors rationale strongly relies on the need to provide such information for non-European ancestries, to which I only partially agree. Yes, we clearly need more non-European research, but not simply for the sake of it. I might be wrong, but I haven't seen any reference in the paper, that would provide a retrospective rationale for the somewhat flawed concept (see below) that two assays that are supposed to measure the same needs to be compared head-to-head at massive scale in different ancestries. That is, there is not a single example that would demonstrate that the correlation between both technologies is different compared to other ancestries. This might have been best done when focusing on PAVs with higher frequency in CKB, but this is missing. Also, there are hundreds of established assays in clinical use that measure proteins and are reliably deployed around the world.

- The rationale of doing such comparative analysis in other ancestries is still not entirely clear to me, why should measurements of the very same proteins by two different technologies differ by ancestry? This might most likely be the case, if there were genetic variants uniquely affecting epitope accessibility, and only in the case of one but not the other technologies targeting this epitope to bind. I cannot see the authors providing any evidence for such effects. This needs to be clearly motivated in the introduction, best with examples. It is really important to be more inclusive in research, but studies should not just be done and motivated for the sake of it, when no apparent distinctions exist, since this otherwise just exaggerates disparities.

- I apologize for being unclear in my previous comment 5 (reviewer 3). What I meant with technical artefact is that the method used to claim 'ancestral-specific' effects (overlapping credible sets) is flawed (see reviewer comments to the paper), and there are indeed no good and robust methods currently available for QTLs in general to help support such claims.

- The study falls still short to provide an improved understanding why so many observational and genetic results between both platforms differ and I do miss some guidance on what researchers that have only one technology at hand may do with their association results, in particular if they do not replicate in other studies using a different technology. Yes, I understand that we than can use a Supplemental table from this paper to say, ok measurements are or are not comparable, but how does this justify a high-impact publication. More generally, how does the study hold up to the promise given in the very last sentence of the abstract. How does this information help, to decide whether to measure Olink or SomaScan in a new study?

-

Manuscript

- Why are there more cis-pQTLs with Olink, but more trans-pQTLs with SomaScan?

- The addition of cis-pQTL investigations is interesting, but somewhat incomplete, since not much else than technical characteristics are used. What about the type of the variant apart from being a PAV? What about overlap with eQTLs that has been repeatedly advocated as means to flag artificial pQTLs.

Response to reviewers' comments

(Ref No.: NCOMMS-23-59940B)

We thank the reviewer for their additional comments. We have revised the manuscript based on the reviewer's and editor's suggestions. The reviewer's verbatim comments are shown below in bold followed by our responses in non-bold text. Page and line numbers indicating where changes have been made refer to the new version of the manuscript. All new changes are highlighted in green. Old changes in previous revisions are marked in yellow and blue.

REVIEWERS' COMMENTS

Reviewer #3

1. Wang et al. did a considerable revision of their work, that improved the overall accessibility of the paper, including some additional interesting observations. However, I still have concerns of what the study adds over and above of what has been published. I clearly see the relevance of providing a comprehensive comparison between both platforms, but this has been done at considerable scale as outlined in my previous comments. Most importantly, the authors rationale strongly relies on the need to provide such information for non-European ancestries, to which I only partially agree. Yes, we clearly need more non-European research, but not simply for the sake of it. I might be wrong, but I haven't seen any reference in the paper, that would provide a retrospective rationale for the somewhat flawed concept (see below) that two assays that are supposed to measure the same needs to be compared head-to-head at massive scale in different ancestries. That is, there is not a single example that would demonstrate that the correlation between both technologies is different compared to other ancestries. This might have been best done when focusing on PAVs with higher frequency in CKB, but this is missing. Also, there are hundreds of established assays in clinical use that measure proteins and are reliably deployed around the world.

Response: We thank the reviewer for the positive note on our additional analyses and substantial improvements made in the revised manuscript. As to this reviewer's continued reservation about the value of our study in a non-European population, we would like to reiterate the fact that comparing Olink and SomaScan platforms in a non-European population is only part of study purpose. Although there have been previous studies that directly compared Olink and SomaScan, our study has substantially expanded from those studies in the three following aspects:

- First, the comparisons in previous studies were made between older versions of Olink and SomaScan that captured smaller number of proteins than those captured in our study. Since proteomic technologies are developing rapidly, both Olink and SomaScan have expanded their assays to a wider range of proteins, and the consistencies of those proteins need to be assessed.
- Second, all previous direct comparisons were made in relatively small samples (<1,600), which could have limited the power of those analyses. Eldjarn et al. (2023) compared Olink pQTLs in UKB and SomaScan pQTLs in deCODE based on large samples, but since the pQTLs were identified in different datasets, other factors such as population differences and sample handling could also explain the differences in their results.
- Third, previous comparisons mainly focused on pQTLs for proteins. To date, no studies have systematically assessed the consistencies of phenotypic associations, or the two platforms' performance in risk prediction for major diseases. Those aspects are particularly important for biomarker discovery and future clinical assay development for translation.

These are important scientific issues to address regardless of whether it is in a European or non-European population. Hence, there was a need to perform a new head-to-head comparison with newer versions of the assay platforms, in a bigger sample, and covering not only pQTLs but also phenotypic associations and risk prediction. Our study is the first such study to address all these

important issues, with a larger sample size and more overlapping proteins than previous studies, as well as involvement of a non-European population. For clarification, we have re-emphasised these points in the revised manuscript.

Regarding pQTLs/PAVs with higher frequencies in CKB, we have included ALDH2 and PLA2G7 as two specific examples. We found that the *cis*-pQTLs for those two proteins colocalised between Olink and SomaScan, which wouldn't be possible to test in non-East-Asian populations due to their low frequencies. We also added an example of PCSK9, whose *cis*-pQTLs were replicated between the platforms in the European population but did not colocalise in CKB. Again, these highlighted the value of studying diverse populations with different lifestyles and genetic architecture.

It is true that established clinical assays measuring plasma proteins (e.g. CRP, IL6) have been used clinically. However, none of those conventional assays have the coverage and throughput to measure thousands of proteins in large numbers of samples using a small amount of blood (20 ul). We believe this is what makes platforms such as Olink and SomaScan unique for discovery research, which could identify new biomarkers and drug targets.

2. The rationale of doing such comparative analysis in other ancestries is still not entirely clear to me, why should measurements of the very same proteins by two different technologies differ by ancestry? This might most likely be the case, if there were genetic variants uniquely affecting epitope accessibility, and only in the case of one but not the other technologies targeting this epitope to bind. I cannot see the authors providing any evidence for such effects. This needs to be clearly motivated in the introduction, best with examples. It is really important to be more inclusive in research, but studies should not just be done and motivated for the sake of it, when no apparent distinctions exist, since this otherwise just exaggerates disparities.

Response: We acknowledge that there has not been much literature comparing proteomic platforms in different ancestry. As discussed in our response above, we believe the strength of the current study is not limited to it being a non-European population. Following the editor's advice, we have removed the part discussing population differences from the introduction and emphasised other points of our rationale (page 4, lines 92-97).

We have also added a new example of PCSK9 on how genetic ancestry might affect platform concordance. Although we identified *cis*-pQTLs for PCSK9 in both Olink and SomaScan, neither of them was found in the European population. Moreover, the *cis*-pQTLs for PCSK9 were replicated between Olink and SomaScan in the European population, but they did not colocalise in CKB. This might indicate the influence of ancestry difference on platform concordance (page 9, lines 206-212; page 16, lines 372-383).

3. I apologize for being unclear in my previous comment 5 (reviewer 3). What I meant with technical artefact is that the method used to claim 'ancestral-specific' effects (overlapping credible sets) is flawed (see reviewer comments to the paper), and there are indeed no good and robust methods currently available for QTLs in general to help support such claims.

Response: We appreciate the reviewer's insights into cross-ancestry pQTL comparison. Indeed, since there have been no optimal methods developed, checking the overlapping credible sets seem to be the most straightforward way to test the transferability of pQTLs. As it is not the focus of the current study, we did not expand on pQTL comparison between ancestries here. This will be further investigated by our future analysis of pQTLs in CKB, UKB, and deCODE.

4. The study falls still short to provide an improved understanding why so many observational and genetic results between both platforms differ and I do miss some guidance on what researchers that have only one technology at hand may do with their association results, in particular if they do not replicate in other studies using a different technology. Yes, I understand that we than can use a Supplemental table from this paper to say, ok measurements are or are not comparable, but how does this justify a high-impact publication. More generally, how does the study hold up to the promise given in the very

last sentence of the abstract. How does this information help, to decide whether to measure Olink or SomaScan in a new study?

Response: We understand the reviewer's critique and hence edited our manuscript (page 2, lines 49-50; page 14, lines 331-332; page 17, lines 405-407; pages 20, lines 492-498). Since we found that each platform has its own strengths, it is difficult to offer one-size-fits-all suggestions on assay selection, especially in the absence of a gold standard. Therefore, the value of our paper for future studies is to offer guidance on study design and results interpretation. Specifically, future studies may consider prioritising proteins that showed high consistencies (observationally and genetically), as those findings are more replicable across platforms and thus more likely to reflect true biological signals.

Manuscript

5. Why are there more *cis*-pQTLs with Olink, but more *trans*-pQTLs with SomaScan?

Response: Indeed, more *cis*-pQTLs were identified for Olink proteins, but the number of *trans*-pQTLs identified was quite similar between Olink and SomaScan (both non-ANML and ANML). We speculate that the difference in *cis*-pQTLs might be due to differences in binding specificity between Olink and SomaScan reagents. Alternatively, Olink reagents might bind more protein isoforms that are influenced by genetic variation. However, without information on the specific sites that Olink/SomaScan reagents bind, we are unable to directly test these hypotheses. We have now discussed this in our revised manuscript (page 14, lines 331-334).

6. The addition of *cis*-pQTL investigations is interesting, but somewhat incomplete, since not much else than technical characteristics are used. What about the type of the variant apart from being a PAV? What about overlap with eQTLs that has been repeatedly advocated as means to flag artificial pQTLs.

Response: We agree with the reviewer that eQTLs are also an important aspect to investigate for potential epitope effects. Indeed, we initially did plan to annotate pQTLs with both PAVs and eQTLs, but we were unable to perform the eQTL analysis due to the lack of eQTL studies available in the East Asian population. Moreover, it would also not be appropriate to check overlap with eQTLs in the European population due to LD differences. Therefore, we have added an additional discussion to the revised manuscript to highlight this as one limitation of our study (page 21, lines 492-498).